# Phasic oxygen dynamics confounds fast choline-sensitive biosensor signals in the brain of behaving rodents

Ricardo M Santos*, Anton Sirota*

Bernstein Center for Computational Neuroscience, Faculty of Medicine, Ludwig-Maximilians Universität München, Planegg-Martinsried, Germany

**Abstract** Cholinergic fast time-scale modulation of cortical physiology is critical for cognition, but direct local measurement of neuromodulators in vivo is challenging. Choline oxidase (ChOx)-based electrochemical biosensors have been used to capture fast cholinergic signals in behaving animals. However, these transients might be biased by local field potential and $O_2$-evoked enzymatic responses. Using a novel Tetrode-based Amperometric ChOx (TACO) sensor, we performed highly sensitive and selective simultaneous measurement of ChOx activity (COA) and $O_2$. In vitro and in vivo experiments, supported by mathematical modeling, revealed that non-steady-state enzyme responses to $O_2$ give rise to phasic COA dynamics. This mechanism accounts for most of COA transients in the hippocampus, including those following locomotion bouts and sharp-wave/ripples. Our results suggest that it is unfeasible to probe phasic cholinergic signals under most behavioral paradigms with current ChOx biosensors. This confound is generalizable to any oxidase-based biosensor, entailing rigorous controls and new biosensor designs.

## Introduction

Acetylcholine (ACh) is an essential modulator of neuronal circuits engaged in high order cognitive operations. During aroused states, high extracellular ACh sets cortico-hippocampal circuits toward memory encoding by enhancing sensory processing, synaptic plasticity and neuronal network rhythmicity (*Hasselmo and McGaughy, 2004*; *Marrosu et al., 1995*; *Steriade, 2004*; *Teles-Grilo Ruivo and Mellor, 2013*). The latter is particularly relevant in the hippocampus, where ACh plays an important role in the processing of episodic and emotional information via modulation of theta oscillations (*Buzsáki, 2002*; *Gu et al., 2017*; *Li et al., 2007*; *Mikulovic et al., 2018*; *Vandecasteele et al., 2014*). Contrarily, low tonic ACh during non-REM sleep permits the occurrence of hippocampal sharp-wave/ripple complexes (SWRs), which are critical for memory consolidation (*Buzsáki, 2015*; *Hasselmo and McGaughy, 2004*; *Norimoto et al., 2012*; *Vandecasteele et al., 2014*).

The current theory on the functional role of ACh in cortical and hippocampal circuits has been mainly derived from brain-state-related correlations of tonic ACh levels with underlying network dynamics, or from strong manipulations of the cholinergic system (*Gu et al., 2017*; *Hasselmo and McGaughy, 2004*; *Li et al., 2007*; *Marrosu et al., 1995*; *Norimoto et al., 2012*; *Vandecasteele et al., 2014*). However, such crude analytic and experimental approaches cannot account for the spontaneous non-stationary interactions between ACh, behavior and neuronal network activity. Recently, fine time-scale measurements of cholinergic activity have provided new insights into this interplay. Fast cholinergic transients in cortical and hippocampal regions have been described in response to sensory sampling, unexpected events, negative reinforcements, and reward-related behavior (*Eggermann et al., 2014*; *Hangya et al., 2015*; *Howe et al., 2017*; *Lovett-Barron et al., 2014*; *Parikh et al., 2007*; *Teles-Grilo Ruivo et al., 2017*). The latter have been captured using electrochemical biosensors in response to detection of cues to rewards and reward

*For correspondence:
santos@bio.lmu.de (RMS);
sirota@bio.lmu.de (AS)

Competing interests: The authors declare that no competing interests exist.

approach or retrieval in freely moving rodents (*Howe et al., 2017*; *Parikh et al., 2007*; *Teles-Grilo Ruivo et al., 2017*). The temporally precise alignment of phasic ACh signals to these events hints for a critical role of ACh on the formation of reward-related memories and on the guiding of learned reward-oriented actions.

The above-mentioned studies highlight the suitability of enzyme-based electrochemical biosensors to capture phasic release of neurotransmitters and neuromodulators (*Chatard et al., 2018*). Additionally, amperometric measurements pick-up currents generated by the local field potential (LFP) (*Santos et al., 2015*; *Viggiano et al., 2012*; *Zhang et al., 2009*), making these sensors ideal for studying the interplay between neuromodulatory tone and neuronal network dynamics. The most successful electrochemical ACh-sensing strategy in vivo has relied on the Choline Oxidase (ChOx)-mediated measurement of extracellular choline (Ch), a product of ACh hydrolysis by acetylcholinesterase. The enzyme catalyzes Ch oxidation in the presence of $O_2$, generating $H_2O_2$, which is oxidized on the electrode surface (*Burmeister et al., 2003*; *Parikh et al., 2004*; *Parikh et al., 2007*; *Teles-Grilo Ruivo et al., 2017*; *Zhang et al., 2010*).

However, despite the apparent success of ChOx-biosensors, the factors that can confound their response in vivo at the fast time-scale, such as LFP-related artifacts and $O_2$-evoked enzyme transients, have not been thoroughly addressed.

Chemical modification of the electrode surface has been a common approach used to effectively reduce electrodes' response to electroactive substances (e.g. ascorbate or dopamine) (*Burmeister et al., 2003*; *Zhang et al., 2010*; *Baker et al., 2015*). Cancellation of neurochemical artifacts generating faradaic currents has been further improved in multi-site sensor designs (*Burmeister et al., 2003*; *Santos et al., 2015*; *Zhang et al., 2010*). By differentially coating the recording sites with matrices that contain or lack the enzyme, sites can be rendered Ch-sensitive or not (sentinel sites), enabling differential measurements with improved selectivity. Although the combination of these two strategies has proven effective on neurochemical artifact removal, the differential coating of the recording sites is not optimal to remove capacitive currents arising from fluctuations in the local field potential (*Zhang et al., 2009*). Although differential measurements are essential to clean biosensor signals (*Santos et al., 2015*; *Zhang et al., 2010*), cross-talk caused by $H_2O_2$ diffusion from enzyme-coated to sentinel sites poses important constraints on the sensor design. The inter-site spacing required to avoid diffusional cross-talk (typically >150 μm, depending on enzyme loadings) leads to uncontrolled differences in the amplitude and phase of LFP across sites, compromising common-mode rejection.

Furthermore, the strategies devised to reduce artifacts that directly generate electrochemical currents (chemical surface modifications or common-mode rejection) and are unrelated to enzymatic response to Ch, do not control for factors influencing immobilized ChOx activity (COA). Given that $O_2$ is a co-substrate of the enzyme, it is crucial to control whether physiological $O_2$ variations can contribute to biosensor responses in vivo leading to distortion of true and detection of false-positive Ch signals. Previous studies have only shown that $O_2$ steady-state responses of ChOx-based biosensors in vitro follow apparent Michaelis-Menten saturation kinetics (*Baker et al., 2017*; *Burmeister et al., 2003*; *Santos et al., 2015*). The relatively narrow linear range of $O_2$-dependent biosensor responses, as compared with estimates of average $O_2$ levels in the brain, has motivated the assumption that ChOx-based biosensors are not affected by in vivo $O_2$ dynamics (*Baker et al., 2017*; *Burmeister et al., 2003*). That might, however, oversimplify the effect of $O_2$ on enzymatic activity since its basal levels and activity-related phasic dynamics widely vary across brain regions and experimental conditions (*Lyons et al., 2016*; *Murr et al., 1994*; *Nair et al., 1987*). Furthermore, previous literature has ignored possible non-steady-state (phasic) biosensor responses to $O_2$, which might arise from local consumption and diffusion of enzyme substrates and reaction products in the sensor coating. These putative transient sensor responses to $O_2$ are particularly relevant as they might temporally overlap with fast cholinergic transients. Therefore, the full assessment of biosensors' $O_2$ dependence requires the characterization of tonic and phasic $O_2$-evoked responses and the simultaneous in vivo measurement of COA (biosensor response) and $O_2$ within the sensor substrate. Yet, when addressed, $O_2$ levels in the tissue have been measured using a separate electrode, often of different geometry and/or having a surface material or modification that differs from the Choline-sensing site (*Baker et al., 2015*; *Dixon et al., 2002*; *Santos et al., 2015*). This approach is therefore prone to bias from heterogeneous tissue $O_2$ dynamics and differential kinetics of electrode

responses to $O_2$. Importantly, these studies have only characterized effect of very slow, tonic changes in $O_2$ levels on sensor response.

Here, we have implemented a novel sensing approach based on differential modification of recording sites' electrocatalytic properties toward $H_2O_2$, resulting in Ch-sensitive and pseudo-sentinel sites. As Ch (or COA) responses depended solely on the intrinsic properties of the metal surface, we could dramatically reduce the size and increase the spatial density of recording sites by using tetrodes as the electrode support. The Tetrode-based Amperometric ChOx (TACO) sensor provides differential responses to changes in COA, interferents and LFP across four bundled 17 μm diameter Pt/Ir wires. Importantly, this multichannel configuration allows highly sensitive measurement of COA and $O_2$ in the same brain spot by using a tetrode site to directly measure the latter. This has not been possible to achieve with conventional enzyme-based biosensors, whose design was constrained by diffusional cross-talk.

We show that the TACO sensor provides a highly selective and sensitive measurement of COA when recording from the brain of behaving animals, effectively suppressing artifacts caused by neurochemicals, LFP and movement. But remarkably, a detailed in vitro characterization and mathematical modeling of biosensors revealed a novel phasic component of sensor's $O_2$ dependence caused by non-stationary enzyme responses to phasic $O_2$ changes. Accordingly, measurements with the TACO sensor in behaving animals revealed fast temporally- and amplitude-correlated $O_2$ and COA dynamics following locomotion bouts and hippocampal SWRs. Causal analysis of this correlation via local or systemic manipulation of $O_2$ dynamics in vivo demonstrated that $O_2$ transients can cause phasic COA responses. Our results demonstrate that the biosensor's phasic $O_2$ dependence causes transient biosensor signals in response to physiological fluctuations in $O_2$. The extent and complexity of $O_2$-related confounds is such that extraction of authentic cholinergic dynamics from the signal is not feasible with currently methodology. Importantly, this $O_2$ ChOx-confounding dynamics is associated with behaviorally and physiologically relevant events and warrants important implications for the interpretation of previous studies relying on ChOx and other oxidase-based sensors as well as for the design of future enzyme-based sensors.

## Results

### The TACO sensor provides a highly selective differential measurement of COA

The TACO sensor is built around Pt/Ir wire tetrode, providing four disc-shaped recording sites with 17 μm diameter in close proximity, resulting in an entire sensor diameter of approximately 60 μm (*Figure 1A*). Such spatial density of recording sites is ideal for common-mode rejection of LFP-related currents and neurochemical dynamics. At this spatial scale, diffusional crosstalk would preclude the use of sentinel sites by differential coatings, as done in conventional biosensors, including our previous ChOx biosensor design (*Burmeister et al., 2003*; *Santos et al., 2015*). Instead, we created *pseudo*-sentinel and Ch-sensing sites by differentially plating the tetrode wires, modifying their electrocatalytic response toward $H_2O_2$. This step was followed by coating the tetrode surface with a common matrix containing ChOx entrapped in chitosan (*Santos et al., 2015*; *Figure 1A*). As for the initial plating steps, all tetrode sites were first mildly plated with gold, which marginally increased the electrode surface area, as inferred from impedances at 1 kHz (419 ± 33 kΩ, n = 28 before vs. 370 ± 16 kΩ, n = 44 after gold plating). Despite the slight decrease in impedance, gold-plating significantly decreased the electrocatalysis of $H_2O_2$ reduction/oxidation at the metal surface. In enzyme-coated electrodes, gold-plated sites exhibited a nearly fivefold smaller $H_2O_2$ sensitivity than an unplated Pt/Ir surface (*Figure 1—figure supplement 1*). Remarkably, following this first step, gold-plated sites could be rendered $H_2O_2$-sensitive upon mild platinization. In enzyme-coated electrodes, there was a nearly 10-fold difference in $H_2O_2$ oxidation currents between Au and Au/Pt sites at 0.4–0.6 V vs. Ag/AgCl (*Figure 1B*). The increase in Au sites' response to $H_2O_2$ above +0.7 V is in agreement with the electrochemical behavior of a pure Au electrode and probably results from the formation of surface oxides (*Burke and Nugent, 1997*; *O'Neill et al., 2004*). Interestingly, the response of Au/Pt electrodes to $H_2O_2$ was higher than that of unplated electrodes (*Figure 1—figure supplement 1*), possibly reflecting an increase in the electrode surface area and/or an electrocatalytic effect caused by the deposition of nanostructured platinum over the gold surface (*Burke and*

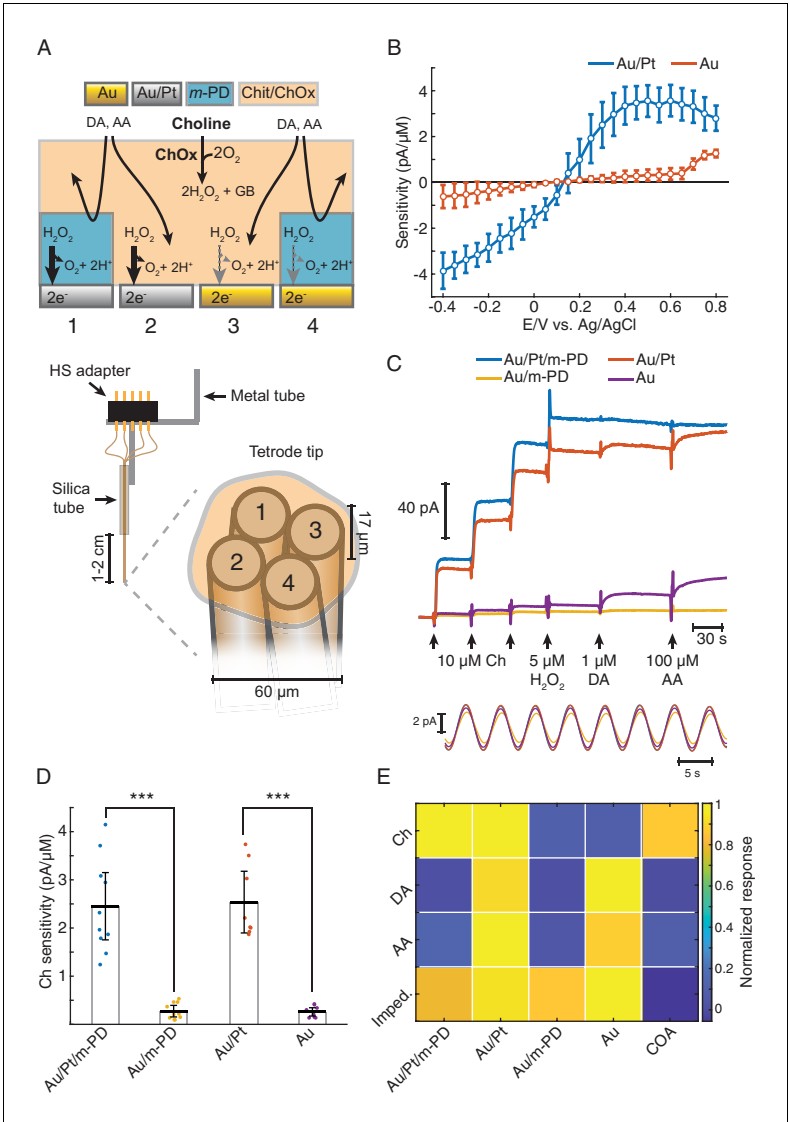

**Figure 1.** TACO sensor design and response properties. (**A**) Schematics depicting the multichannel biosensor design, including the assembly used in in vitro and head-fixed recordings (bottom). HS: head-stage. (**B**) Voltammogram showing $H_2O_2$ sensitivities of gold-plated and platinized sites upon amperometric calibrations at different DC potentials (n = 10). Prior to calibrations, tetrodes were coated with a matrix of chitosan/ChOx. (**C**) Top shows a representative calibration of a sensor showing the response of different types of sites to step additions of Ch, $H_2O_2$, dopamine (DA). and ascorbate (AA). Bottom shows an example of current responses to a sinusoidal 12 mV AC voltage at 0.2 Hz overlaid on top of +0.6 V vs. Ag/AgCl DC voltage. (**D**) Sensitivities of different sites toward Ch (n = 10 biosensors). Unlike m-PD electropolymerization, platinization significantly increased sensitivity (p<0.0001 and $F_{1,33}$ = 115 for platinization effect and p=0.87 and $F_{1,33}$ = 0.03 for m-PD, by two-way ANOVA for unbalanced data). (**E**) Normalized responses of each tetrode site to Ch, interferent molecules and AC voltage, presented as impedance at 0.2 Hz (n = 5–10). Magnitudes significantly depended on the site modification and on the factor tested (p<0.0001 and $F_{1,113}$ = 22 for platinization, $F_{1,113}$ = 109 for m-PD and $F_{3,113}$ = 12.9 for factors, by three-way ANOVA for unbalanced data). Platinization selectively increased responses to Ch (p<0.0001) while m-PD decreased responses only to DA and AA (p<0.0001). Impedances did not significantly differ across different types of electrode modifications (p>0.99). The rightmost column shows COA signal responses computed from the difference between Au/Pt/m-PD and Au/m-PD sites. Groups were compared by three-way ANOVA followed by Tukey-Kramer post-hoc tests. Data are represented as mean ± CI.

The online version of this article includes the following figure supplement(s) for figure 1:

**Figure supplement 1.** Effect of platings on platinum electrode sensitivity to $H_2O_2$.

*Buckley, 1996*; *Domínguez-Domínguez et al., 2008*). Even when using common-mode rejection, decreasing the magnitude of interferences in individual sites is desirable. Thus, after plating and coating the tetrode, we electropolymerized *m*-PD in two of the recording sites, in order to reduce responses to electroactive compounds larger than $H_2O_2$ (e.g. ascorbate or dopamine) (*Hascup et al., 2013*; *Santos et al., 2008*). The final TACO sensor configuration consisted of all possible combinations of Pt and *m*-PD modifications of Au-plated recording sites (*Figure 1A*).

The responses of TACO sensors' sites to Ch, $H_2O_2$ and to compounds that can potentially interfere during in vivo measurements were tested by step additions in the beaker at +0.6V vs. Ag/AgCl (*Figure 1C*, top). In accordance with the voltammograms of $H_2O_2$ sensitivities (*Figure 1B*), the responses of TACO sensors' gold-plated sites to Ch and $H_2O_2$ were much lower than those of platinized sites. On average, like for $H_2O_2$, this difference was about 10-fold for Ch, regardless of *m*-PD electropolymerization (*Figure 1D*). Contrasting with its lack of effect on Ch sensitivity, *m*-PD dramatically decreased responses to ascorbate and dopamine, regardless of the site's metal composition (*Figure 1E*). In addition, we have also calibrated the impedance of recording sites at low frequency by applying a low-amplitude 0.2 Hz AC voltage on top of the DC offset. This low frequency is of particular relevance, as it overlaps with putative phasic cholinergic dynamics previously reported by us and other groups in anesthetized and freely moving rodents (*Howe et al., 2017*; *Parikh et al., 2007*; *Santos et al., 2015*; *Teles-Grilo Ruivo et al., 2017*). Impedances calculated from the current oscillations generated by the AC voltage (*Figure 1C*, bottom) were comparable across all sites (*Figure 1E*). Collectively, the results summarized in *Figure 1E* and *Table 1* validate the gold-plating approach to produce *pseudo*-sentinel sites, as it selectively reduces electrode's response to $H_2O_2$. The COA signal most consistently used throughout this study was computed from the differential of *m*-PD-electropolymerized sites, to exploit the advantages of both differential platings and *m*-PD electropolymerization, resulting in a high selectivity for Ch (or changes in COA) (*Figure 1E* and *Table 1*). As compared to our previous stereotrode design using 50 µm diameter wires, these sensors keep the same Ch response performance, with a limit of detection (LOD) in the low nanomolar range, remarkable for such small electrode surfaces. The TACO sensor response is stable, without significant drop upon in vivo head-fixed recordings (2.45 ± 1.11 pA/µM before and 2.46 ± 0.80 pA/µM after implantation, n = 8, p=0.99, paired *t*-test), shows high linearity within the physiological range and a $T_{50}$ response time around 1.5 s (*Table 1*). Noteworthy, though the response of our sensors is expected to be mostly shaped by Ch diffusion in the coating (*Santos et al., 2015*), the presented response times are in fact slightly overestimated due to a delay caused by mixing of the analyte in the stirred calibration buffer. Additionally, while *m*-PD abrogates the response of electropolymerized

**Table 1.** Analytical properties of TACO sensors.

| | **Individual sites' analytical properties** | | | | | |
|---|---|---|---|---|---|---|
| **Channel type** | **Ch sensitivity (pA/µM)** | **Ch sensitivity (nA µM$^{-1}$ cm$^{-2}$)** | **$H_2O_2$ sensitivity (pA/µM)** | **DA sensitivity (pA/µM)** | **AA sensitivity (pA/µM)** | **Impedance (GΩ)** |
| Au/Pt/*m*-PD | 2.45 ± 0.70 (n = 10) | 1081 ± 309 (n = 10) | 2.48 ± 0.93 (n = 8) | 0.15 ± 0.23 (n = 10) | 0.02 ± 0.013 (n = 10) | 2.90 ± 0.48 (n = 5) |
| Au/Pt | 2.54 ± 0.64 (n = 8) | 1118 ± 283 (n = 8) | 2.86 ± 1.10 (n = 6) | 7.48 ± 1.86 (n = 8) | 0.22 ± 0.20 (n = 8) | 3.38 ± 0.56 (n = 5) |
| Au/*m*-PD | 0.27 ± 0.12 (n = 10) | 118.2 ± 53 (n = 10) | 0.22 ± 0.16 (n = 8) | 0.13 ± 0.23 (n = 10) | 0.007 ± 0.004 (n = 10) | 3.11 ± 0.77 (n = 5) |
| Au | 0.25 ± 0.086 (n = 9) | 111.5 ± 38 (n = 10) | 0.29 ± 0.08 (n = 7) | 8.24 ± 1.84 (n = 9) | 0.19 ± 0.16 (n = 9) | 3.65 ± 0.59 (n = 5) |

| | **Analytical properties for COA measurement (Au/Pt/*m*-PD - Au/*m*-PD)** | | | | |
|---|---|---|---|---|---|
| **Ch sensitivity (pA/µM)** | **Limit of detection (nM)** | **Linearity, [Ch]<30 µM (R$^2$)** | **Response time (s)** | **DA sensitivity (pA/µM), selectivity ratio** | **AA sensitivity (pA/µM), selectivity ratio** |
| 2.18 ± 0.73 (n = 10) | 28 ± 0.011 (n = 10) | 0.9996 ± 0.0003 (n = 10) | 1.4 ± 0.4 (n = 10) | 0.022 ± 0.38 (n = 10), 101:1 | 0.013 ± 0.012 (n = 10), 169:1 |

The data are given as the mean ± CI (95%).

The number of sensors tested is given in parentheses. Data were collected from calibrations on the day after *m*-PD electropolymerization.

sites to large electroactive molecules, the differential site modifications provide further information on the signal identity. As the Au site (without *m*-PD) is more responsive to interferents than to Ch, the neurochemical confounds (NCC) signal (please see Materials and methods section) enables to further infer contaminations by neurochemical confounds in vivo. The differential sites' responses to different factors can potentially be further exploited by multivariate methods of analysis, and the electrochemical tetrode design employed in TACO sensor can be generalized to other types of sensors in the future.

## Freely moving recordings validate the TACO sensor suppression of current-generating artifacts and suggest a potential $O_2$ modulation of COA in vivo

In order to test whether the TACO sensor can measure fast COA transients devoid of artifacts that directly generate electrode currents in behaving animals (regardless of a possibly underlying $O_2$ modulation of COA), we have first performed recordings in freely moving animals. We simultaneously recoded from the CA1 pyramidal layer using the TACO sensor and LFP across hippocampal depth using a 32-channel linear silicon probe implanted in the proximity of the biosensor (*Figure 2A*). Extracellular electrophysiology allowed us to directly validate the amperometric measurement of high-frequency LFP (*Figure 2—figure supplement 1*). The COA signal was cleaned by subtraction of the *m*-PD-electropolymerized *pseudo*-sentinel from the Ch-sensing sites' signal, upon frequency-domain correction of the *pseudo*-sentinel amplitude and phase (please see Materials and methods) (*Santos et al., 2015*). This procedure led to substantial removal of fast current fluctuations ascribed to LFP (example recording in *Figure 2B*, top). Accordingly, the median spectral power at ~1–20 Hz of the signal derived from the Au/Pt/m-PD site during both NREM sleep and wake periods decreased more than two orders of magnitude after the signal cleaning procedure (*Figure 2B*, bottom). The cleaned COA signal tonically changed across different brain states reaching higher values during active wakefulness and REM sleep than during NREM sleep (*Figure 2—figure supplement 2*). This dynamics is compatible with expected brain state-dependent ACh changes (*Hasselmo and McGaughy, 2004*; *Marrosu et al., 1995*), but remarkably, COA also fluctuated on the time-scale of seconds within brain states.

Hippocampal SWRs and arousal-related locomotion bouts are discrete events associated with major phasic changes in hippocampal network activity, occurring during sleep or wakefulness respectively (*Buzsáki, 2002*; *Buzsáki, 2015*; *Sirota et al., 2003*). Thus, we tested whether these events could correlate with phasic COA dynamics. Hippocampal SWRs were detected from a silicon probe site in the CA1 pyramidal layer during NREM sleep. Average raw biosensor signals triggered on the peak of SWRs showed a prominent peak in all TACO sensor's sites (*Figure 2C*, top). The similarity of peak amplitudes suggests an LFP-related origin of these currents, which was virtually absent in the cleaned signals (Figure, 2C, middle), and reflects the slow dynamics of the sharp wave. The high magnitude of this LFP artifact emphasizes the importance of the common-mode rejection approach in revealing COA dynamics that would otherwise be masked. Interestingly, regardless of *m*-PD electropolymerization, the cleaned COA signal showed a peak lagging the SWR by ~3 s. Importantly, the slow and small amplitude dip in the neurochemical confounds signal excludes the role of interferents (e.g. ascorbate or a monoamine) in the COA transient (*Figure 2C*, middle).

Bouts in locomotion, detected as peaks in the rat running speed, were associated with transient increases in theta power (*Figure 2D*, bottom) and with a transient current deflection in all TACO sensor sites (*Figure 2D*, top). Slow time scale currents that might have neuronal, muscle or movement artifact origin were cleaned, similar to SWRs, revealing a slow peak in COA signals following locomotion bouts, but not in the neurochemical confound signal assuring negligible contribution of these interferences in the COA signal (*Figure 2D* middle).

These results highlight the usefulness of our multichannel *pseudo*-sentinel approach to discriminate between authentic changes in COA and different sorts of current-inducing interferents, including LFP- and movement-related artifacts and neurochemical dynamics. Nevertheless, it is noteworthy that phasic changes in COA were associated with the most salient sources of non-stationarity in hippocampus, namely arousal/locomotion and SWRs (*Buzsáki, 2002*; *Buzsáki, 2015*; *Sirota et al., 2003*). These phenomena can potentially correlate with changes in both ChOx substrates, Ch (*Norimoto et al., 2012*; *Vandecasteele et al., 2014*; *Hasselmo and McGaughy, 2004*; *Marrosu et al., 1995*; *Reimer et al., 2016*; *Teles-Grilo Ruivo and Mellor, 2013*) and $O_2$

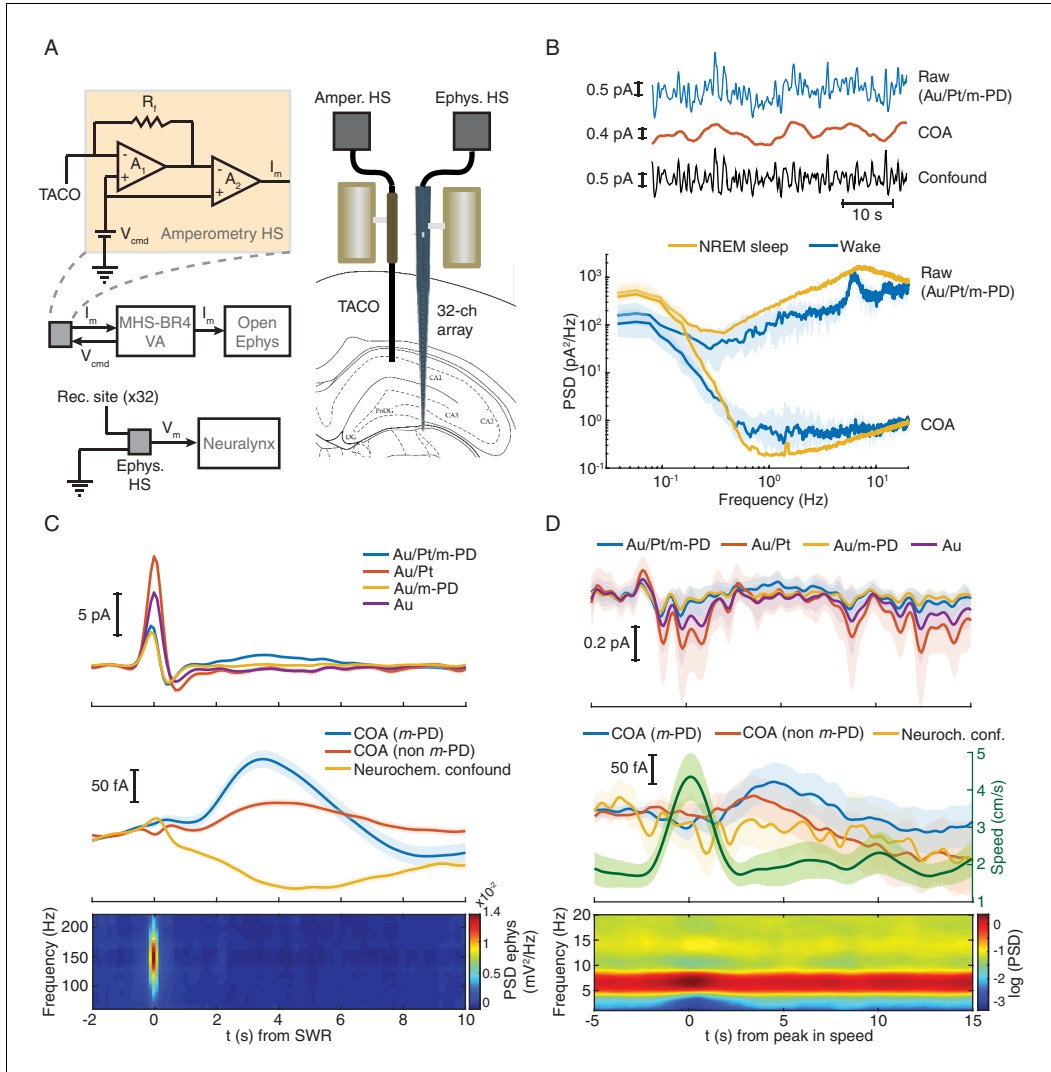

**Figure 2.** TACO sensor provides a highly sensitive and selective measurement of COA in freely behaving animals. (**A**) Left panel shows a simplified diagram of the electronic circuits in freely moving recordings. The amperometric measurement (one channel for simplicity) was based on a voltage clamp circuit. $R_f$: feedback resistor (1 GOhm); $A_1$, $A_2$: OP amps; $V_{cmd}$: command voltage; $I_m$: measured current (acquired as analog voltage signal). $V_{cmd}$ is set on the MHS-BR4-VA box and sent to the head-stage. Its output is acquired by an Open-Ephys system. Electrophysiological setup is depicted in bottom. $V_m$: measured voltage output. The amplified LFP signal is acquired by the Neuralynx system. Right panel depicts the arrangement of a TACO sensor and a 32-channel silicon probe chronically implanted in the hippocampus of a rat. Both probes were attached to microdrives. (**B**) Top, segment of NREM sleep recording, a raw signal (low-pass filtered at 1 Hz), cleaned COA and confound components. Bottom, spectrum of the raw and clean COA signal during wake (n = 10) and NREM sleep periods (n = 19). Data shown as medians ± CI. (**C**) Average low frequency (1 Hz low-pass filtered) biosensor signals and high frequency power spectrograms triggered to SWRs detected from a silicon probe channel in CA1 pyramidal layer (top quartile of all ripples sorted by power, n = 2019). Average raw (top), cleaned (middle) COA responses and LFP spectrogram triggered on SWRs (bottom). (**D**) Average raw (top) cleaned COA responses (middle), average speed (middle) and LFP spectrogram (bottom) triggered to peaks in rat speed in an open-field arena (n = 127). Data are represented as mean ± CI, except in B.

The online version of this article includes the following figure supplement(s) for figure 2:

**Figure supplement 1.** Amperometric currents reliably track LFP spectral content over a wide frequency range.

**Figure supplement 2.** Tonic COA dynamics across brain states in a freely moving animal.

(*Zhang et al., 2019*; *Ramirez-Villegas et al., 2015*), leading to transient modulation of COA. Thus, our freely moving results further emphasized the need for a detailed investigation of the $O_2$ effect on the COA signal.

## Biosensor's COA responses to oxygen have tonic and phasic components

Since $O_2$ is a co-substrate of oxidases, physiological $O_2$ variations potentially affect the response of oxidase-based biosensors. However, this issue has not been thoroughly addressed in previous studies (*Baker et al., 2015*; *Chatard et al., 2018*; *Dixon et al., 2002*; *McMahon et al., 2007*; *Santos et al., 2015*). Here, we sought a detailed investigation of the biosensor $O_2$-dependence in vitro, enabling the assessment of $O_2$ effect on COA at multiple time scales (i.e. tonic and phasic dynamics). In these experiments, we took advantage of the TACO sensor configuration to sample COA and $O_2$ dynamics from the same spot, necessary for accurate evaluation of $O_2$-dependence of the COA signal. These recordings were performed using a new customized head-stage allowing independent control of the potential applied on each recording site. Oxygen measurement was achieved at a negative potential by $O_2$ reduction on a gold-plated site, which was typically maximal at −0.4 V vs. Ag/AgCl (*Figure 3A*). In these experiments, clean COA and $O_2$ signals were obtained by subtraction of Au/Pt/$m$-PD (at +0.6 V) and Au (at −0.2 V) by the *pseudo*-sentinel $m$-PD site, respectively.

The in vitro tests were based on step additions of known $O_2$ concentrations in the presence of a background Ch concentration (5 µM) representative of average brain extracellular Ch tonic levels (*Brehm et al., 1987*; *Garguilo and Michael, 1996*; *Parikh et al., 2004*). Importantly, unlike previous studies, this allowed us to distinguish and probe the contribution of phasic and steady-state (tonic) COA responses to $O_2$.

Notably, upon removal of $O_2$ from solution and in the presence of background Ch, most biosensors responded to consecutive $O_2$ steps with a fast transient (phasic component) before reaching a steady-state (*Figure 3B*). Both phasic and tonic components decreased with $O_2$ baseline, but not equally. The phasic response was usually still prominent upon the exhaustion of the tonic component (*Figure 3B*). We quantified these differences by fitting the Michaelis-Menten equation to the tonic changes and the Hill equation to the cumulative phasic peaks, as the latter did not appear to follow a pure Michaelis-Menten profile (*Figure 3C*). The resulting phasic $K_{0.5}O_2$ values for phasic responses were, on average, one order of magnitude larger than tonic $KmO_2$, reinforcing that the phasic component vanishes at much larger $O_2$ baselines than tonic responses (p<0.0005, *Figure 3D*). Importantly, the phasic ChOx response was not matched by a fast $O_2$ transient following each addition, as $O_2$ raised considerably slower than the peak in COA (*Figure 3E*).

Next, we assessed how the coating composition and its physical properties could modulate the sensor $O_2$-dependence. First, we calculated the biosensor efficiency (ratio of Choline vs. $H_2O_2$ sensitivities), as a proxy to the enzyme loading in the coating and plotted it against tonic $KmO_2^{app}$ (*Figure 3F*). The result revealed a decreasing trend (*Figure 3F*), suggesting that biosensors with a high enzyme loading have low sensitivity to tonic $O_2$ changes (low-$KmO_2^{app}$). Strikingly, the biosensors with the lowest tonic $KmO_2^{app}$ exhibited the highest phasic peaks (*Figure 3—figure supplement 2*), with maximal phasic responses to $O_2$ decreasing exponentially as a function of $KmO_2^{app}$ (*Figure 3G*).

In order to further detail on the effect of $O_2$ baseline on non-stationary COA, we split the sensor calibrations into two groups according to their tonic $O_2$-dependence. As already suggested in *Figure 3E*, we confirmed that the larger phasic responses in the low-$KmO_2^{app}$ vs. high-$KmO_2^{app}$ groups could not be attributed to differences in the underlying $O_2$ dynamics. Oxygen transients after each addition were negligible and did not significantly differ across $KmO_2^{app}$ groups (p=0.998, *Figure 3H*). Noteworthy, in the low-$KmO_2^{app}$ group, the highest COA peak was achieved in response to the second $O_2$ addition (10 µM of cumulative [$O_2$]) rather than to the first in six out of seven calibrations (marginally significant difference between 5 and 10 µM $O_2$ responses, p=0.076, paired $t$-test). Furthermore, the same COA peaks had the longest decay across the two $KmO_2^{app}$ groups (p<0.0005, *Figure 3I*). These non-monotonic profiles with respect to [$O_2$] reflected the deviation of the cumulative phasic COA vs. $O_2$ curves from a Michaelis-Menten kinetics. Accordingly, the Hill coefficients extracted from calibration fittings (e.g. *Figure 3C*) were above two for sensors with low $KmO_2^{app}$ and significantly decreased toward one as $KmO_2^{app}$ increased (p<0.05, *Figure 3J*). These

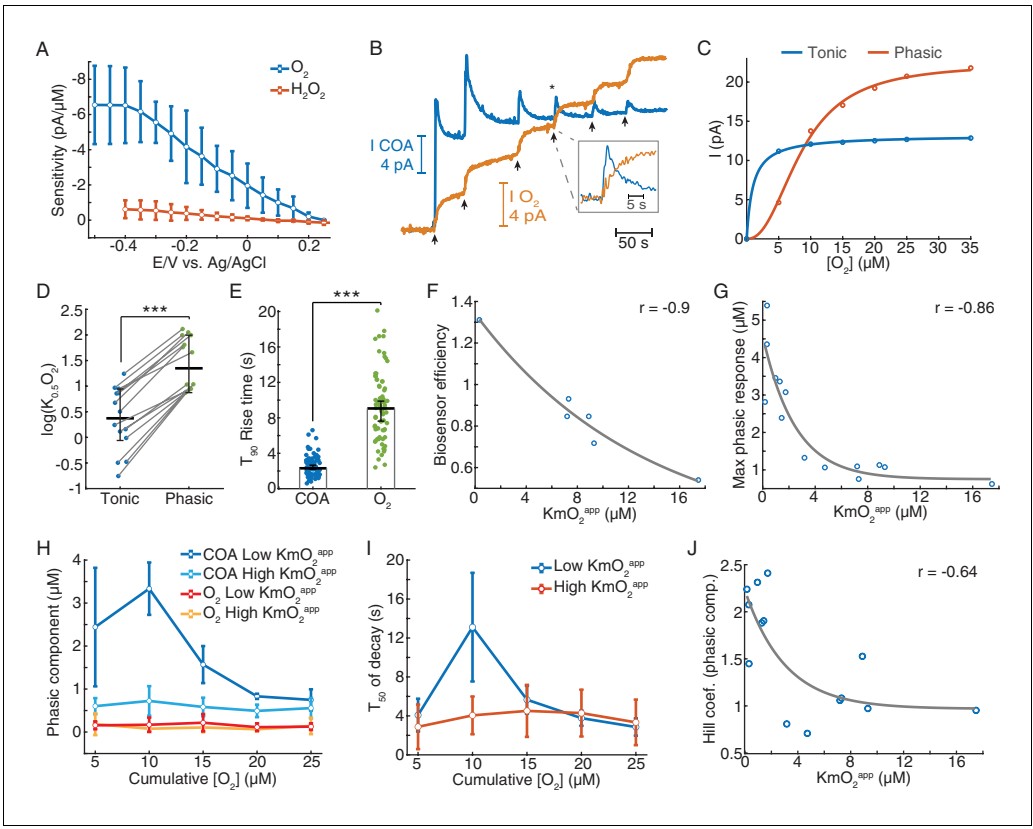

**Figure 3.** Biosensors generate tonic and phasic COA components in response to $O_2$. (**A**) The voltammogram shows the DC voltage-related sensitivity of gold-plated sites toward $H_2O_2$ (n = 10) and $O_2$ (n = 5). (**B**) An example of an in vitro $O_2$ calibration. Upon removal of $O_2$ from PBS containing 5 µM Ch, 5 µM $O_2$ additions (arrows) were performed until the tonic sensor response saturated. (**C**) Representative cumulative tonic and phasic responses as a function of $O_2$ baseline after each addition. Tonic data were fitted with Michaelis-Menten equation, resulting in a $KmO_2^{app}$ of 0.97 µM (CI = 0.818–1.126 µM) and $I_{max}$ = 13.2 pA (CI = 13.04–13.37 pA), with RMSE = 0.08 pA. Phasic responses were fitted to the Hill equation, yielding $K_{0.5}O_2$ = 8.6 µM (CI = 7.58–9.62 µM), Hill coefficient $n$ = 2.31 (CI = 1.74–2.88) and $I_{max}$ = 22.4 pA (CI = 20.6–24.1), with rmse = 0.50 pA. (**D**) $K_{0.5}O_2^{app}$ values of tonic and phasic components from all biosensors (n = 14). Averages and error bars are medians and CIs. Groups were significantly different (sign test, p<0.0005). (**E**) $T_{90}$ rise times of $O_2$ steps and COA peaks following $O_2$ additions (n = 72–95). Bars and error bars represent medians and 95% CIs, groups were significantly different (p<0.0001, Wilcoxon rank sum test). (**F**) Biosensor efficiency (Ch/$H_2O_2$ sensitivity ratio) as a function of $KmO_2^{app}$ (n = 6). For illustrative purposes, data were fitted with an exponential function. Spearman correlation between variables was −0.9 (p=0.028). (**G**) Maximal phasic response to $O_2$ (from each biosensor calibration) as a function of $KmO_2^{app}$ (n = 14, $r_{spearman}$ = −0.86, p<0.0001). Data were fitted with an exponential decay curve (decay constant $k$ = 0.4 µM$^{-1}$, CI = 0.031–0.78 µM$^{-1}$, rmse = 0.68 µM). (**H**) Amplitudes of phasic ChOx responses divided into low and high-$KmO_2^{app}$ groups (n = 7 per group) following $O_2$ additions. Control phasic $O_2$ using the same algorithm is plotted for the same groups (n = 6 for high-$KmO_2^{app}$ and n = 4 for low-$KmO_2^{app}$). Low-$KmO_2^{app}$ COA transients were higher than those from the high-$KmO_2^{app}$ group (p<0.0001) and the amplitudes from both COA groups were higher than any $O_2$ group (p<0.005). Oxygen transients from low-$KmO_2^{app}$ vs. high-$KmO_2^{app}$ groups did not significantly differ (p=0.998). Group comparisons by two-way ANOVA for unbalanced data followed by Tukey-Kramer post-hoc tests. Data are means ± CI. (**I**) $T_{50}$ decay of transients from low- and high-$KmO_2^{app}$ groups as a function of cumulative $O_2$. The decay of peaks at 10 µM $O_2$ was the longest (p<0.0005, two-way ANOVA for unbalanced data followed by Tukey-Kramer post-hoc tests). Data are means ± CI. (**J**) Hill coefficient from the fits of cumulative COA phasic component vs. cumulative $O_2$ (as in C) as a function of $KmO_2^{app}$ (n = 14). The two variables negatively correlate ($r_{spearman}$ = −0.64, p=0.015), showing an exponential-like relationship (fitted initial amplitude of 1.27 ± 0.75, decay constant of 0.33 ± 0.62 µM$^{-1}$ and offset of 0.97 ± 0.67). ***p<0.001.

The online version of this article includes the following figure supplement(s) for figure 3:

**Figure supplement 1.** Oxygen measurement with the TACO sensor does not affect the simultaneously recorded COA signal.

**Figure supplement 2.** Phasic biosensor responses to $O_2$ in vitro.

observations suggest a cooperative mechanism that enhances phasic responses as the $O_2$ baseline increases, in biosensors with low tonic $O_2$ dependence.

In summary, we show that, under a physiological Ch background, ChOx biosensors respond to $O_2$ with a transient increase in enzyme activity before reaching a steady-state. Importantly, phasic and tonic components were apparently mutually exclusive and their relative magnitude was sensitive to the properties of the enzyme coating, namely enzyme loading. Sensors with high enzyme loading have a low tonic $O_2$-dependence but show large phasic responses whose amplitude is modulated by the $O_2$ baseline.

## Modeling in vitro biosensor responses reveals the mechanisms underlying tonic and phasic oxygen-dependence

In order to provide a theoretical ground for our in vitro observations and further understand the mechanisms underlying tonic and phasic COA responses to $O_2$, we have simulated the behavior of biosensors in calibration conditions. We numerically solved a system of partial differential equations describing the diffusion of Ch and $O_2$ in the coating and their interaction with the enzyme, leading to $H_2O_2$ generation.

To mimic our experimental calibrations, we simulated sensor responses to 5 µM step increases in $O_2$ (starting from zero) under a constant level of 5 µM Ch in the bulk solution. Remarkably, in line with our experimental findings, the model predicted phasic and tonic components of sensor response to $O_2$ whose magnitude depended on $O_2$ baseline (*Figure 4A*). Sensors with high enzyme loading showed higher phasic peaks and tonic responses that saturate at lower $O_2$ baselines than sensors with a low enzyme amount (*Figure 4A*). To get a more resolved characterization of biosensor's $O_2$ dependence, we next generated response curves at 1 µM $O_2$ steps, until saturation of enzymatic $H_2O_2$ generation was nearly reached (see Materials and methods). By simulating a range of coating thicknesses and enzyme concentrations, we found that both parameters decreased $KmO_2^{app}$ of tonic responses (*Figure 4B*) and increased the magnitude of phasic peaks (*Figure 4C*). Interestingly, our model predicts that particular combinations of coating thickness and enzyme concentration can be used to optimize sensor sensitivity to Ch (at saturating $O_2$ levels) (*Figure 4—figure supplement 1A*). Yet, such a strategy is not expected to concomitantly reduce phasic and tonic $O_2$ dependence, which seem to be mutually exclusive. In agreement with the experimental data, our model anticipated that sensors with the lowest $KmO_2^{app}$ exhibit the highest transient responses to $O_2$ (*Figure 4B–D*). Across the simulated coatings, the highest phasic peaks occurred when sensor cumulative tonic responses were close to maximal and progressively decreased in sensor tonic saturation level as a function of tonic $KmO_2^{app}$ (*Figure 4D*). This observation is in agreement with our experimental observation of highest phasic peaks at a 10 µM $O_2$ baseline (*Figure 3H–J*).

To get further clues into the factors shaping sensors' $O_2$-evoked responses, we analyzed the concentration dynamics of Ch and $O_2$ in the coatings during simulated calibrations. We observed that, under high enzyme loading, Ch is rapidly depleted in the coating as $O_2$ levels in solution increase (*Figure 4E*). This effect is less pronounced in sensors with low enzyme loading (*Figure 4F*). Interestingly, significant $O_2$ consumption, observed mainly in coatings highly loaded with enzyme, was stronger for low $O_2$ levels before reaching saturation of the sensor tonic response (*Figure 4G*). These observations suggest that depletion of Ch in the enzyme coating is the limiting factor that shapes sensors' tonic responses to $O_2$.

To further investigate phasic responses under high enzyme loading, in addition to substrate dynamics, we assessed the levels enzyme-bound reduced intermediate species vs. buffer $O_2$ concentration, as those are direct precursors of $H_2O_2$. We observed that, under non-saturating $O_2$ levels, the concentrations of $E_{red}BA$ and $E_{red}GB$ change oppositely as $O_2$ is increased, which maintains the sum of both intermediates relatively stable (*Figure 4—figure supplement 1B*). At 1 µM buffer $O_2$ steps, the profiles of reduced intermediates, Ch and $O_2$ at the electrode surface, suggest that phasic biosensor peaks result from a combination of multiple factors (*Figure 4H*). As expected, the sensor's tonic response steeply decays upon Ch depletion in the coating. This decrease is accompanied by a sharp rise in the instantaneous $\Delta O_2$ at the electrode surface evoked by $O_2$ increments in the bulk solution. As the rate of $O_2$ consumption depends on the concentration of reduced enzyme-bound complexes, the increasing profile of $\Delta O_2$ results, in a first stage, from the depletion of the $E_{red}BA$ complex, followed by the decrease in $E_{red}GB$ (*Figure 4—figure supplement 1B*). In turn, the amount of $E_{red}BA$ and $E_{red}GB$ depends on both Ch and $O_2$, which leads to a sum that is shifted to the right

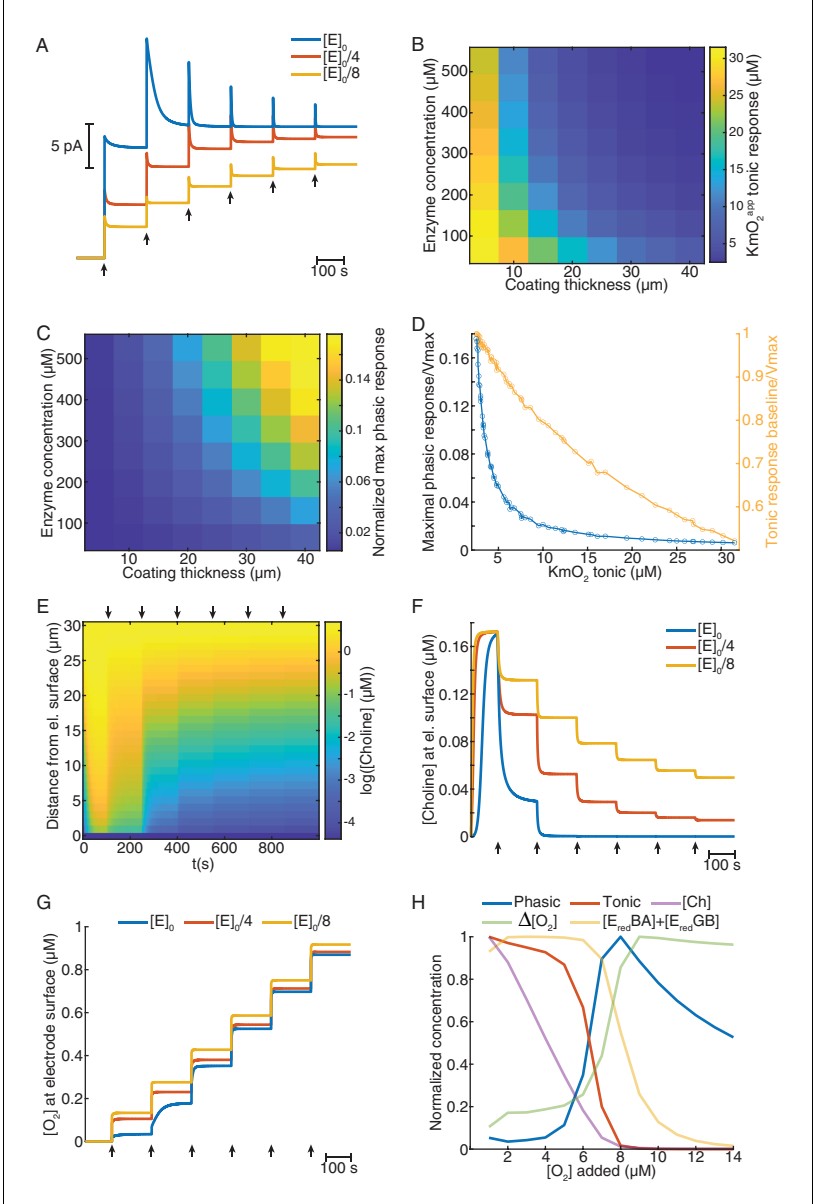

**Figure 4.** Mathematical model explains ChOx-based biosensor COA responses to oxygen. (**A**) Simulated calibrations of biosensors with different enzyme concentrations in the coating (coating thickness of 30 μm). Choline in bulk solution was kept constant at 5 μM and $O_2$ was incremented in 5 μM steps from zero to 30 μM (arrows). (**B**) $KmO_2^{app}$ of tonic responses as a function of coating thickness and enzyme concentration in the coating. (**C**) Normalized maximal phasic responses of sensors with different coating thicknesses and enzyme concentrations. Magnitudes refer to the highest phasic response divided by the maximal cumulative tonic response (Imax) from the respective simulated sensor calibration. (**D**) Blue trace represents the normalized maximal phasic response vs. tonic $KmO_2^{app}$ and the orange trace refers to the level of saturation of the sensor's tonic response at which the maximal phasic peak occurs. Data were obtained from all combinations of coating thickness and enzyme concentration in B and C. (**E**) Concentration dynamics of Ch in the sensor coating as a function of distance from the electrode surface during a simulated calibration of a high enzyme-loaded sensor (coating thickness of 30 μm and enzyme concentration of 560 μM, same as the blue in A). Arrows indicate 5 μM $O_2$ step increments in solution. (**F**) and (**G**) show time-courses of Ch and $O_2$ concentration, respectively, at the electrode surface of sensors with different enzyme concentrations during simulated calibrations. Arrows indicate 5 μM $O_2$ increments in solution. (**H**) Normalized profiles of phasic and tonic sensor responses as well as of concentrations of enzyme substrates and total reduced enzyme-bound complexes at the electrode surface as a function of $O_2$ in solution. Data are from a simulated calibration of a sensor with a 30 μm coating and an enzyme concentration of 560 μM, upon 1 μM $O_2$

*Figure 4 continued on next page*

*Figure 4 continued*
step increases in solution. The $\Delta O_2$ is the initial rise in $O_2$ following each $O_2$ increment in solution (at a lag of 0.3 s).
The online version of this article includes the following figure supplement(s) for figure 4:

**Figure supplement 1.** Mathematical simulation of ChOx-biosensor tonic responses and reduced enzyme-bound intermediate dynamics.
**Figure supplement 2.** Minimal kinetic mechanism of Choline Oxidase.

relative to the sensor's tonic response. Thus, phasic sensor peaks reach maximal levels at the offset of tonic responses, when there is a combination of relatively large concentrations of the direct $H_2O_2$ precursors, namely enzyme-bound reduced complexes and $O_2$, and very low levels of Ch in the coating (*Figure 4H*).

Overall, our biosensor simulations corroborate the in vitro results, firstly converging toward the notion that Ch consumption in the coating is a major factor governing sensor tonic $O_2$ dependence. Secondly, the results suggest that non-steady state phasic responses depend on the relative concentrations of Ch, reduced intermediate enzyme complexes and $O_2$ in the coating. This biochemical mechanism is expected to convert physiological $O_2$ changes into spurious COA phasic signals and calls for careful investigation of the $O_2$ modulation of COA dynamics in vivo.

## Phasic COA during locomotion bouts correlates with oxygen transients

The TACO sensor offers the opportunity to probe the activity of immobilized ChOx, as a putative index of cholinergic tone, with unprecedented spatial resolution and selectivity in behaving animals. However, our in vitro data and mathematical simulations hint toward a possible confounding effect of a phasic biosensor $O_2$-dependence on fast time-scale measurements of Ch in vivo. To directly address this question in the brain of behaving rodents, we have exploited the advantages of the tetrode configuration, ideally suited to measure multiple electroactive compounds at the same brain spot. In vivo recordings were performed using the same electrode configuration as previously described in vitro (*Figure 5A*, top). In this case, clean COA and $O_2$ signals were obtained by frequency-domain-corrected subtraction of Au/Pt/*m*-PD (at +0.6 V) and Au (at −0.2 V) by the *pseudo*-sentinel *m*-PD site, respectively, as described in Materials and methods section.

In order to obtain more controlled experimental conditions and overcome technical constraints posed by the size of the head-stage, the remaining experiments were performed in head-fixed mice spontaneously running on a treadmill (*Figure 5A* bottom). Strikingly, we found that phasic COA dynamics typically matched the simultaneously recorded $O_2$ fluctuations, which were generally related to changes in behavioral state (*Figure 5B*). In line with the freely moving data (*Figure 2D*), head-fixed running bouts were temporally correlated with an increase in the power of theta oscillations as well as with delayed phasic increases in both COA and $O_2$, peaking a few seconds later (*Figure 5B and C*). This lag was significantly greater than zero in all recording sessions (one-sample *t*-test, p<0.01), averaging $3.85 \pm 2.04$ s (n = 5) for COA and $5.04 \pm 3.12$ s (n = 5) for $O_2$. Notably, running-related isolated COA or $O_2$ peaks were rare, with the vast majority of the events showing either no identifiable change or co-occurrence of the two transients (*Figure 5D*). Amplitudes of COA peaks were significantly correlated with those of $O_2$ (p<0.001) when pooling together the events from all recordings, which allowed sampling across the full $O_2$ amplitude range (*Figure 5E*). Phasic COA correlated more consistently with $O_2$ than with theta power or speed (*Figure 5F*). Moreover, running-bout-related COA and $O_2$ peak lags were significantly correlated (p<0.001, *Figure 5G*) and, interestingly, within most sessions (three out of five), COA peaked significantly earlier than $O_2$ (p=0.029, p<0.0001 and p<0.0001 for significant sessions, paired *t*-test).

In summary, these results indicate a strong correlation between phasic COA and $O_2$ in the hippocampus of head-fixed mice following locomotion bouts.

## Phasic COA and oxygen signals follow clusters of sharp-wave/ripples during immobility

Hippocampal SWRs are critical for memory consolidation and their occurrence has been proposed to be anti-correlated with cholinergic activity in the hippocampus (*Hasselmo and McGaughy, 2004*;

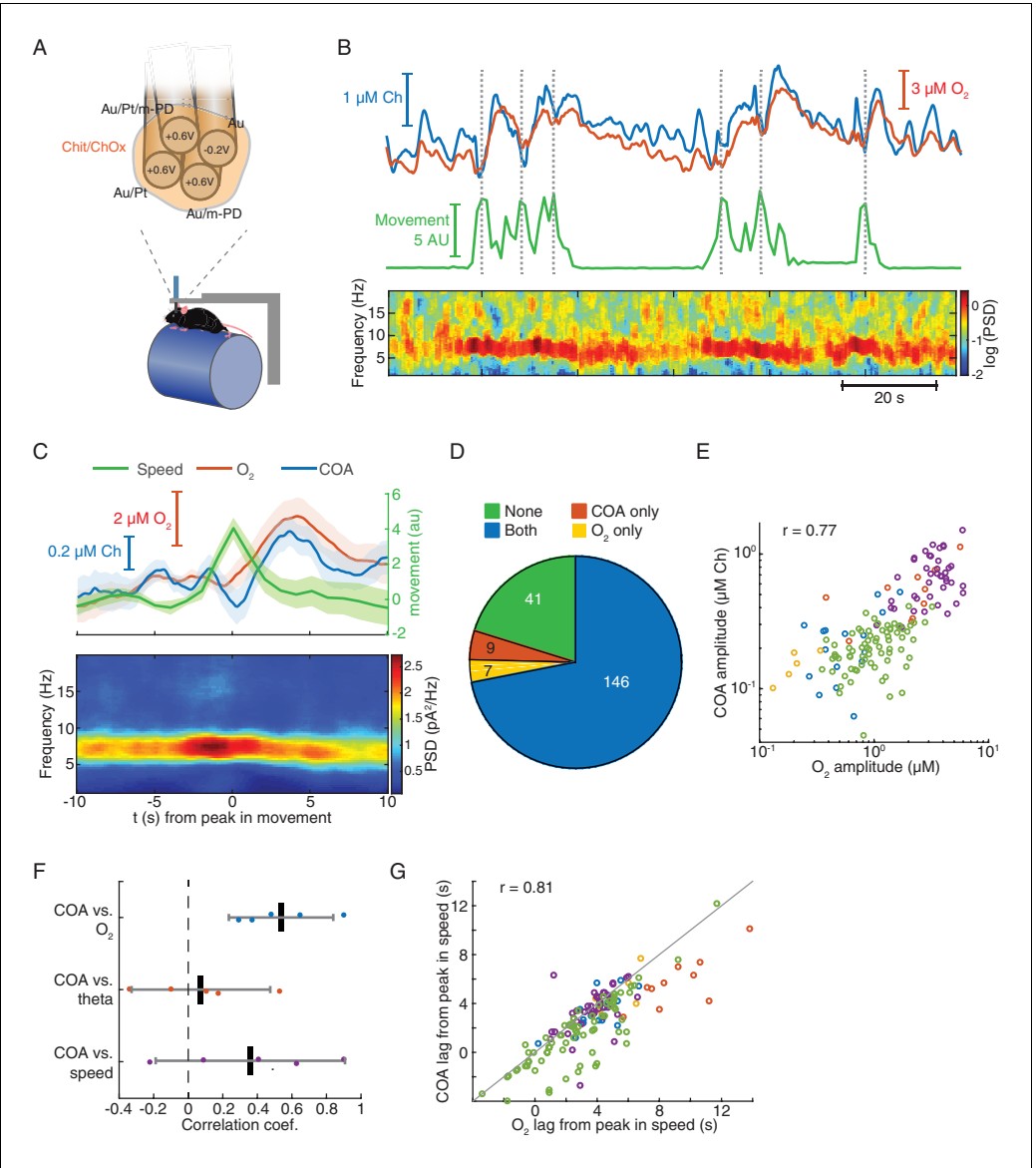

**Figure 5.** Locomotion-related correlated changes in COA and local oxygen concentration in head-fixed mice. (**A**) The schematics of the tetrode configuration used to simultaneously measure COA ($H_2O_2$) and extracellular $O_2$.in head-fixed mice. Values on each recording site indicate the applied DC voltage *vs.* Ag/AgCl (+0.6 V for $H_2O_2$ and −0.2 V for $O_2$ measurement). (**B**) Representative segment of simultaneous recording of COA and $O_2$ (top), locomotion speed (middle) and LFP spectrogram (bottom). Dashed lines indicate times of detected locomotion bouts. (**C**) Average speed, COA and $O_2$ signals (top) and LFP power spectrogram (bottom) triggered on locomotion bouts (n = 41 from one recording session). (**D**) Event counts across different categories of locomotion bouts, distinguished by the occurrence or absence of COA and/or $O_2$ peaks. Data were collected from five recordings in three mice. (**E**) Amplitude of COA (shown as calibrated Ch concentrations) vs $O_2$ transients following locomotion bouts (data from the events with increases in both signals, n = 146). Each color represents one recording session (n = 5–74 events per recording). Amplitudes were significantly correlated ($r_{spearman}$ = 0.77, p<0.001). (**F**) Spearman correlation coefficients between COA amplitudes and $O_2$, theta power or speed (n = 5). Correlation across recordings was significant for ChOx *vs* $O_2$ data (p<0.01, *t*-test). Each point represents a single recording session. (**G**) Peak lags of COA *vs.* $O_2$ relative to the peak in speed ($r_{spearman}$ = 0.81, p<0.001, n = 146). Colors represent recording sessions (n = 5–74 events per recording). The diagonal line was plotted to ease comparison between lags. Data were obtained from five recordings in three mice and are represented as mean ± CI.

*Norimoto et al., 2012; Vandecasteele et al., 2014*). However, our freely moving data showing COA transients following SWRs contradicts this prediction, posing questions on the factors driving the biosensor response during these events. Thus, we investigated whether the SWR-related response of immobilized ChOx was correlated with extracellular $O_2$ in head-fixed mice during periods of quiescence.

Remarkably, on average, SWR events were followed by fast transients in both COA and $O_2$ (*Figure 6A*). Both COA and $O_2$ peak amplitudes correlated best with ripple power integrated over a period of ~2 s lagging them by 3–4 s (*Figure 6B and C*). Similarly, both SWR count and summed ripple power integrated in a 2 s window positively correlated with the delayed amplitude of both COA and $O_2$ transients (*Figure 6D–E* and *Figure 6—figure supplement 1*).

These findings might indicate the contribution of a time-constant related to the sensor response and/or to a relatively slow physiological process by which SWRs recruit cholinergic activity or a local hemodynamic response leading to $O_2$ increase. Indeed, functional MRI has reported SWR-triggered increases in BOLD signal in the primate hippocampus, reflecting a local tissue hemodynamic response at a time-scale matching $O_2$ transients observed here (*Ramirez-Villegas et al., 2015*). Importantly, the amplitudes of SWR-associated COA and $O_2$ phasic transients were consistently correlated within all recordings (n = 3, p<0.001, *Figure 6F*). Similarly, lags of these transients to SWR were significantly correlated (*Figure 6G,H,I*).

Together, the data indicate correlated phasic changes in COA and extracellular $O_2$ in response to SWRs (*Figure 6I*), especially when they happen in clusters. As in the case of locomotion bouts, at this stage one could not rule out the contribution of neither phasic Ch nor $O_2$ as the trigger for the peaks in COA. The putative cholinergic origin of this response would, nevertheless, be surprising given the suppressive effect of ACh on SWR occurrence (*Norimoto et al., 2012; Vandecasteele et al., 2014*). Thus, in light of the $O_2$ transients that accompany the rise in COA and of our in vitro findings of the phasic $O_2$ dependence in this type of biosensors, these observations cast doubt on the validity of the putative SWR-triggered cholinergic response.

## Interactions between COA and oxygen are not sensitive to ongoing hippocampal dynamics and depend on the time-scale

The correlation between COA and $O_2$ following locomotion and SWRs may reflect interaction between the two signals or result from the coincident recruitment of cholinergic and hemodynamic responses. While a consistent relationship between the two variables is expected in the first case, irrespective of ongoing network dynamics, the same may not happen in the latter. To get insights into this question, we analyzed an additional category of events, consisting of fast $O_2$ transients detected outside the time-windows surrounding SWRs and peaks in locomotion. Remarkably, virtually all (>95%) of these events had an associated COA transient. The dynamics of COA and $O_2$ were typically similar (*Figure 7A*), with COA peaks lagging, on average, from −0.09 s to 0.48 s (n = 3 recordings) relative to $O_2$. Signal amplitudes were significantly correlated both when pooling together all events (p<0.0001, *Figure 7B*) or within all individual sessions (p<0.006). The amplitude correlation coefficients were in the range of those obtained for $O_2$ peaks associated with locomotion and SWRs (*Figure 7C*), but the average amplitudes of $O_2$ and corresponding COA signals were in the sub-micromolar range, comparable to those associated with SWRs (*Figure 7D*). Overall, although the amplitude range of locomotion-related peaks was wider, it is notable that the whole data fit to a COA/$O_2$ relationship that seems to follow the same model across recordings and event types (*Figure 7D*). These observations are therefore compatible with an interaction between COA and $O_2$, although contribution of third-party factors could not be definitely excluded at this stage.

Besides the amplitude of $O_2$ transients, the temporal profile of $O_2$ rise might influence the shape and amplitude of associated ChOx responses and thus provide further hints on the causality and directionality of COA-$O_2$ interactions. This analysis is of particular relevance in light of the phasic component of biosensor's $O_2$ dependence that we have uncovered by our in vitro tests and mathematical modeling. In vivo we detected spontaneous $O_2$ peaks at different frequency bands (ignoring their correlation with SWRs or speed), resulting in a spectrum of $O_2$ rise times from less than 2 to 14 s (please see Materials and methods section for details). Across all recording sessions, we consistently observed that the COA peak anticipated $O_2$ (negative lag) as $O_2$ rise time increased (*Figure 7E,F* and *Figure 7—figure supplement 1A*). Maximal COA lags averaged 0.38 ± 0.42 s and were associated with fast $O_2$ rises, lasting 1.4–2.6 s (*Figure 7F and G*). Furthermore, the slope of

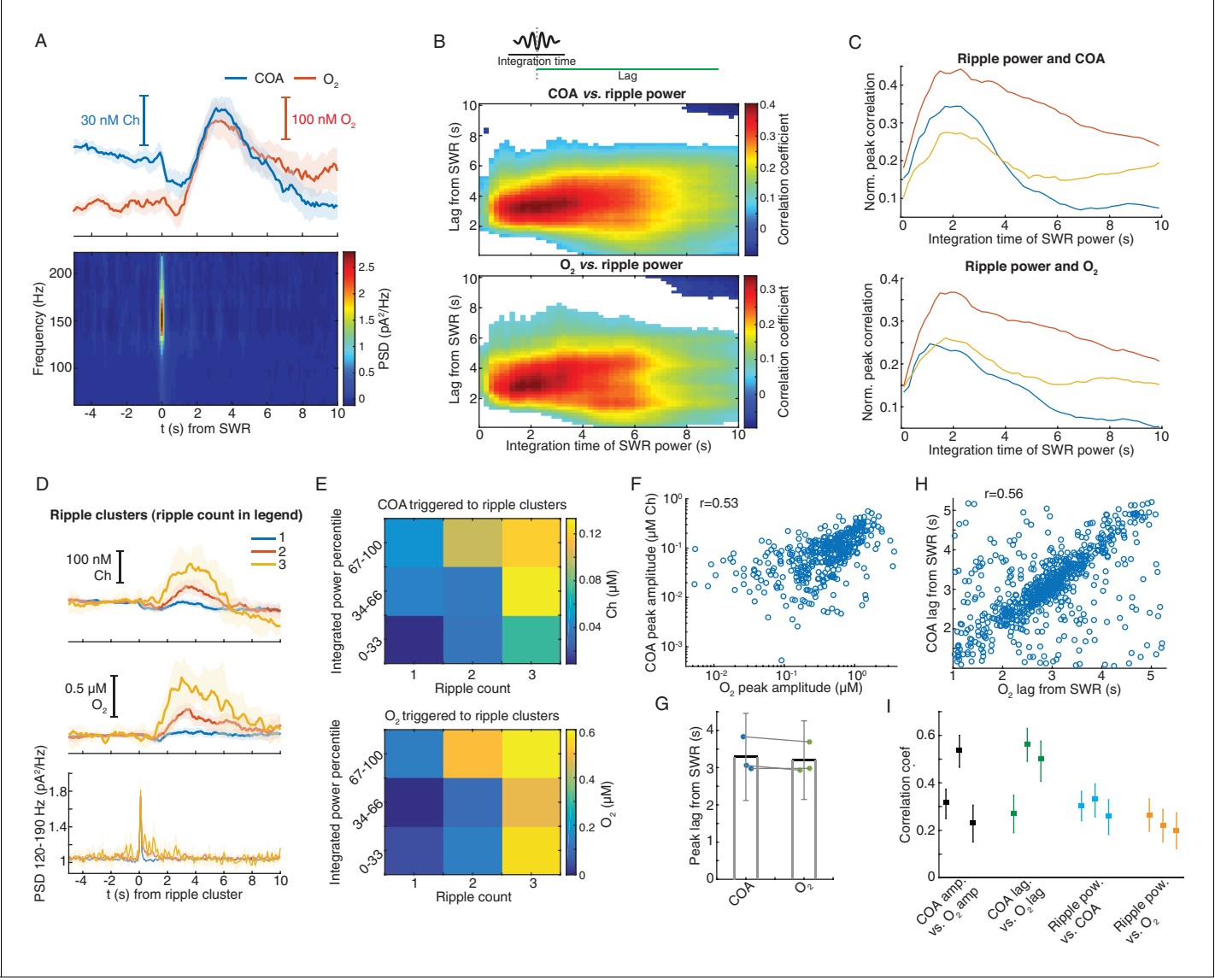

**Figure 6.** Correlated COA and oxygen transients follow hippocampal SWRs. (**A**) Average COA and $O_2$ dynamics triggered on hippocampal SWRs (n = 1067, top) and average LFP spectrogram (bottom) from a recording session in the head-fixed setup. (**B**) Pseudo-color-coded Spearman correlation between COA (top) or $O_2$ (bottom) amplitude at different time lags from SWRs (y-axis) and integrated ripple band power computed in windows of varying size (x-axis) for a representative session. White areas represent non-significant correlations (p>0.05). (**C**) Normalized peak correlations, obtained from the difference between maximal and minimal correlations for each integration time in B, between integrated ripple power and COA or $O_2$. Colors represent recording sessions in different mice. (**D**) Average COA, $O_2$ and ripple power dynamics triggered to SWRs bursts containing variable number of SWRs (1-3) in a 2 s window (n = 34–591 SWRs in each group, respectively from one representative session). Data are represented as median ± CI. (**E**) Peak amplitude of COA or $O_2$ transients as a function of SWR burst size (as shown in D) sorted by different percentile ranges of summed ripple power. Both SWR count and total ripple power significantly affected the amplitude of COA and $O_2$ (two-way ANOVA for unbalanced data following ART, p<0.0001 and $F_{2,803}$ > 12 for both factors in COA and $O_2$ data). (**F**) Amplitude of COA vs. $O_2$ transients following SWRs from one recording session ($r_{spearman}$ = 0.53, p<0.0001, n = 1067). (**G**) Group statistics on the lags of COA and $O_2$ peaks relative to SWRs. Each dot is the average from one recording. Bars represent means ± CI. (**H**) Lags of COA peaks as a function of $O_2$ peak lags relative to SWRs. The parameters were significantly correlated ($r_{spearman}$ = 0.56, p<0.0001, n = 1067). (**I**) Summary of correlations between sensor signals and ripples. Each point represents one recording session, and error bars are CIs computed using bootstrap. Data were obtained from three recording sessions in three mice.

The online version of this article includes the following figure supplement(s) for figure 6:

**Figure supplement 1.** Phasic COA and oxygen responses following SWRs jointly depend on their power and grouping.

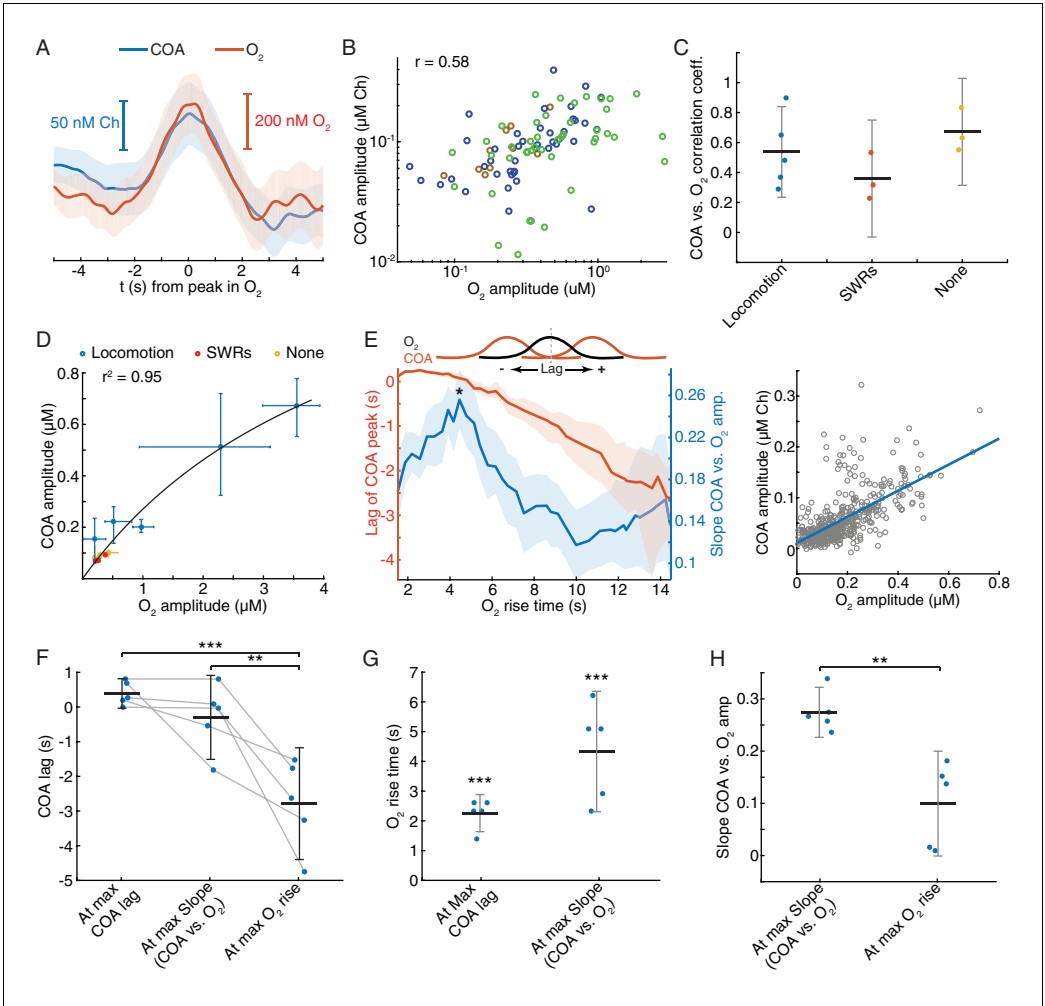

**Figure 7.** Correlation between COA and spontaneous $O_2$ transients suggests $O_2$-COA directionality. (**A**) Average COA and $O_2$ dynamics triggered to fast $O_2$ transients (duration ~5 s) detected outside periods when SWRs or locomotion bouts occurred. Data is from one recording session (n = 42 events). (**B**) Relationship between amplitudes of $O_2$ and associated COA transients outside SWRs/locomotion bouts. Colors represent different recording sessions (n = 10–45 from three recordings, $r_{spearman}$ = 0.58, p<0.0001). (**C**) Group summary of COA *vs.* $O_2$ amplitude correlations under different behavioral and/or electrophysiological contexts. (**D**) COA vs. $O_2$ amplitude across animals and states. Each point represents the median of events from a recording session (error bars are CIs). Data were fitted to the Michaelis-Menten equation (Vmax = 1.58, Km = 4.84). (**E**) Lag of COA peaks relative to $O_2$ peaks (red) and slope of COA *vs.* $O_2$ amplitude (blue) for $O_2$ transients with varying rise time for a single session; medians with CIs (left). Right shows the relationship between amplitudes of $O_2$ and associated COA transients for the $O_2$ rise time corresponding to largest slope marked with * on the left panel. (**F**) Group statistics on maximal COA lags, lags at maximal COA/$O_2$ slope and lags associated with longest $O_2$ transients (ANOVA, $F_{2,12}$ = 15.22, with post-hoc Tukey test, p=0.51 for max COA lag vs. lag at max COA/$O_2$, p=0.0006 for max COA lag vs. lag at max $O_2$ rise and p=0.0038 for lag at maximal COA/$O_2$ slope vs. lag at max $O_2$ rise). (**G**) Group statistics on $O_2$ rise times corresponding to maximal COA lags and COA/O2 slope in (**E**). Values from both groups were significantly lower than the longest $O_2$ rise time observed in a given recording (p<0.0001, one-sample t-test). (**H**) Group statistics on the slopes of COA vs. $O_2$ amplitude at its maximum value and at maximum $O_2$ rise. Differences between groups were significant (p=0.002, paired t-test). ***p<0.001, **p<0.01. Data were obtained from five recording sessions in three mice and are represented as mean ± CI, except in D.

The online version of this article includes the following figure supplement(s) for figure 7:

**Figure supplement 1.** COA dynamics associated with $O_2$ transients across all experiments.

COA vs. $O_2$ amplitudes was time-scale dependent, peaking for $O_2$ transients that took 2.3 to 6.2 s to rise (*Figure 7E,G* and *Figure 7—figure supplement 1B*) and progressively decreasing for longer $O_2$ rises (*Figure 7E,H* and *Figure 7—figure supplement 1B*). The time-scale dependence of the COA vs. $O_2$ slope is apparently related to the non-linear interaction between COA and $O_2$, as a function of $O_2$ rise time. Contrasting with the approximately linear increase in $O_2$ amplitude as a function of $O_2$ rise (*Figure 7—figure supplement 1C*), the amplitude of corresponding COA peaks was, on average, nearly time-scale independent for $O_2$ rise times in the range of 3–14 s (*Figure 7—figure supplement 1D*).

These results provide important insights into the interaction between COA and extracellular $O_2$ in the hippocampus. The advancement of COA relative to $O_2$ could be compatible with a Ch-$O_2$ directionality, possibly caused by ACh-evoked changes in local blood flow (*Takata et al., 2013*). However, the diffusional delay underlying such a mechanism is expected to be constant as a function of time-scale. Moreover, the time-scale dependence of the relative amplitude of enzyme response is hard to interpret in light of a Ch-$O_2$ directionality. In contrast, these in vivo observations are fully compatible with the phasic component of biosensor's $O_2$ dependence that we have found in vitro and supported by a mathematical model of the sensor. In agreement with in vivo interplay between COA and $O_2$, the discovered phasic $O_2$ dependence predicts that both tonic (or slow) and phasic changes in $O_2$ would evoke a transient increase in COA preceding the peak in $O_2$. Such modulation is expected to be time-scale-dependent, being maximal at relatively short time-windows. Additionally, in line with in vivo COA and $O_2$ dynamics, this phenomenon predicts a continuous drift in COA-$O_2$ peak lags as $O_2$ rise time increases.

Together, these results converge to the hypothesis that the observed changes in COA are mainly caused by phasic non-steady-state responses of ChOx to $O_2$ fluctuations in vivo.

## Exogenous oxygen transients in the hippocampus elicit phasic COA responses

Our correlational analysis in vivo, supported by in vitro characterization of biosensor $O_2$ dependence and mathematical modeling of the biosensor, points toward an effect of $O_2$ transients on phasic COA when recording from the hippocampus. We tested the causality of this interaction first by evoking changes in $O_2$ by local application of small volumes of $O_2$-saturated saline through a glass micropipette, positioned at a few hundred microns from the biosensor tip. To evoke different $O_2$ dynamics, we varied ejection parameters such as time and pressure.

Remarkably, immobilized ChOx exhibited robust phasic responses to exogenous $O_2$ transients, regardless of the time-scale of $O_2$ change (*Figure 8A* and *Figure 8—figure supplement 1A*). Amplitudes of COA and $O_2$ were significantly correlated (p<0.0001, *Figure 8B*). However, similarly to spontaneous dynamics, the COA/$O_2$ amplitude ratio of single events (equivalent to COA/$O_2$ slope in spontaneous data) significantly decreased with $O_2$ rise time in all experiments (negative correlation, p<0.05, *Figure 8C*). The decrease in COA amplitude was accompanied by the advancement of its peak relative to $O_2$ (p<0.001 in all recordings, *Figure 8D*). Thus, these results qualitatively recapitulate our observations for the spontaneous COA-$O_2$ interactions, reinforcing the putative $O_2$-COA directionality.

Next, we manipulated hippocampal $O_2$ levels non-invasively, through inhalation, to further confirm $O_2$ causality. We exposed mice to a pure $O_2$ stream during variable periods (4–30 s) in order to generate different $O_2$ transients. Like in the case of local $O_2$ delivery, we observed reproducible changes in COA in response to exogenous $O_2$ (*Figure 8E* and *Figure 8—figure supplement 1*). The data showed a significant correlation between peak amplitudes (p<0.0001, *Figure 8F*). Importantly, both the COA/$O_2$ amplitude ratio and COA peak lag significantly decreased as a function of $O_2$ rise time (p<0.05), corroborating the conclusions from local $O_2$ manipulation.

Overall, $O_2$ inhalation tended to generate slower and smaller $O_2$ transients (and perhaps more physiological) than those by local application. Despite the differences in magnitudes and time-scales, the tested correlations are consistent across the two paradigms (*Figure 8I*) as well as with the spontaneous interactions between COA and $O_2$ (*Figure 7E–H*). Both spontaneous and evoked $O_2$-COA interactions are therefore fully compatible with our in vitro characterization of a novel phasic $O_2$-dependence of this type of sensors caused by non-steady-state enzyme activity in response to $O_2$. Our results converge to the conclusion that the putative cholinergic transients observed in the

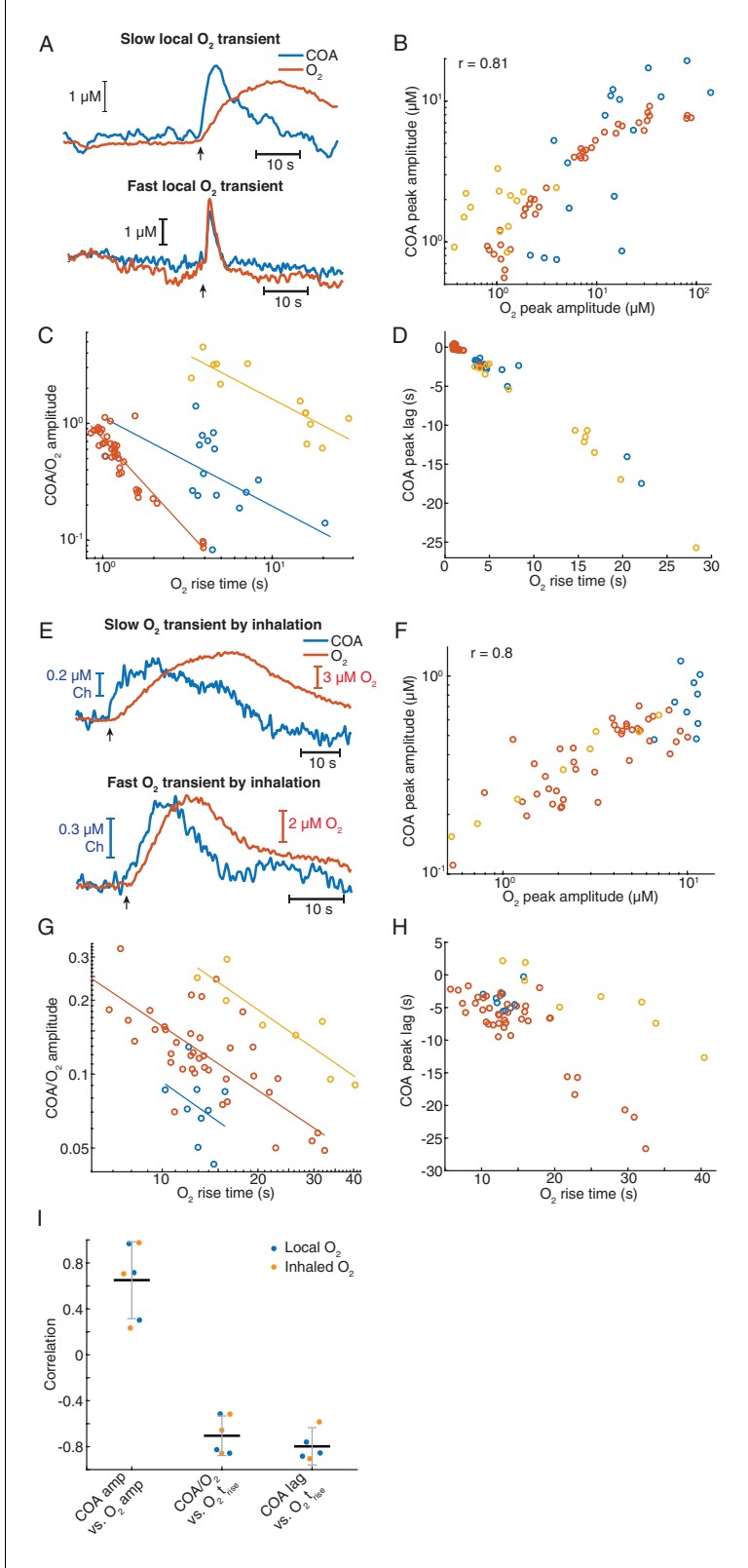

**Figure 8.** Exogeneous $O_2$ elicits phasic COA responses in the hippocampus in vivo. (**A**) Representative examples of slow and fast $O_2$ transients and associated COA responses evoked by local application of exogenous $O_2$ from a glass micropipette. (**B**) Amplitudes of COA vs. locally evoked $O_2$ transients. Data are from three recordings (color-coded, n = 13–40 per session). Amplitudes were significantly correlated when pooling all data ($r_{spearman}$ = 0.81,

*Figure 8 continued on next page*

*Figure 8 continued*

p<0.0001) and within two sessions (p=0.0017 and p<0.0001, analysis not performed in one session due to the narrow range of $O_2$ amplitudes covered). (**C**) Ratio of COA *vs.* evoked $O_2$ peak amplitudes as a function of $O_2$ signal rise time. Colors denote recording sessions and trendlines are linear fits performed on data from each session. In all cases, the variables were negatively correlated (p=0.037, p<0.0001, and p<0.001 for each session). (**D**) Lag of COA relative to locally evoked $O_2$ transient peaks as a function of $O_2$ rise time. Lags significantly decreased with $O_2$ rise time in all sessions ($r_{spearman}$ for each recording ranged from −0.76 to −0.88, p<0.001). (**E**) Representative traces showing slow and fast transients evoked by $O_2$ inhalation. E-H plots are analogous to A-D, but for $O_2$-inhalation-induced $O_2$ transients. (**F**) Amplitudes of COA *vs.* $O_2$ transients. Data were obtained from three recordings (color-coded, n = 8–40 per session). Whole data as well as data from individual experiments were significantly correlated (p<0.0001 and p<0.0005, respectively). (**G**) Amplitude ratio of COA *vs.* $O_2$ transients as a function of $O_2$ rise time. Variables were negatively correlated (red and yellow sessions, with $r_{spearman}$ of −0.66 and −0.86 respectively, p<0.0001 and p=0.011). (**H**) Lags of COA relative to $O_2$ peaks significantly decreased as a function of $O_2$ rise time (red and yellow sessions, $r_{spearman}$ of −0.58 and −0.90 respectively, p<0.0001 and p=0.0046). Analysis in F-H not performed in one session (blue dots) due to insufficient coverage of $O_2$ amplitudes and rise times. (**I**) Summary of amplitude and COA-$O_2$ lags correlations for COA and $O_2$ transients evoked by local $O_2$ application and $O_2$ inhalation. Data were obtained from six recording sessions in five mice and are presented as means ± CIs.

The online version of this article includes the following figure supplement(s) for figure 8:

**Figure supplement 1.** Raw sensor responses evoked by exogenous oxygen in vivo.

---

hippocampus of behaving rodents rather result from phasic modulation of COA by physiological $O_2$ fluctuations.

## Discussion

We have developed a novel multi-site tetrode-based amperometric ChOx (TACO) sensor optimized for the highly sensitive and unbiased simultaneous measurement of immobilized ChOx activity (COA), as an index of cholinergic activity, and $O_2$ dynamics in the brain. Our approach, based on the differential plating of recording sites to create *pseudo*-sentinel channels outperforms previous common-mode rejection strategies, which were limited by large distance between sensor and sentinel sites, impedance mismatch and diffusional cross-talk (*Burmeister et al., 2003*; *Parikh et al., 2007*; *Santos et al., 2015*). This strategy allowed us to substantially reduce the size and increase the spatial confinement of recording sites by using a 17 µm wire tetrode as the biosensor electrode support. Our recordings in freely moving and head-fixed rodents reveal the usefulness of this compact multi-site design to clean artifacts from sensor signals and assess the correlation between the activity of the immobilized enzyme and brain extracellular $O_2$ on a fast time-scale. Importantly, this method can be generalized to improve the selectivity and address the in vivo $O_2$-dependence of any oxidase-based biosensor.

By simultaneously measuring COA and extracellular $O_2$ in the hippocampus of behaving mice, we found that fast biosensor signals correlate in amplitude and time with $O_2$ transients evoked by behavioral and network dynamics events exemplified by locomotion bouts and SWRs. Notably, the relationship between COA and $O_2$ dynamics was apparently not sensitive to the underlying neurophysiological or behavioral context and was preserved during periods without appreciable SWR incidence or locomotion. By using two different methods to manipulate extracellular $O_2$, we show that $O_2$ fluctuations in the physiological range can drive phasic COA. Remarkably, the time-scale dependence of biosensor response amplitude and lag relative to exogenous $O_2$ qualitatively matched that of spontaneous dynamics, suggesting that the same directionality takes place in the spontaneous conditions.

Locomotion-related $O_2$ elevations in head-fixed mice have been recently shown to be modulated mainly by respiration rate (*Zhang et al., 2019*), whereas SWR-evoked $O_2$ peaks have been indirectly inferred by fMRI and likely result from neurovascular coupling (*Leithner and Royl, 2014*; *Ramirez-Villegas et al., 2015*). Thus, our study provides a link between the neurophysiological or systemic mechanisms that modulate brain $O_2$ levels and the response of ChOx-based biosensors in vivo. As it is an intrinsic component of any behavioral task, our results highlight the importance of controlling for $O_2$-evoked biosensor signals related to locomotion or movement. It is thus likely that, in reward-

related tasks, locomotion related to reward retrieval elicits few seconds delayed phasic changes in $O_2$ that drive transient increases in COA. Likewise, a high incidence of SWRs in reward locations, reflecting a consummatory state (*Buzsáki, 2015*), might trigger $O_2$ transients and, in turn, phasic ChOx biosensor responses. These two examples provide alternative explanations for previously reported cholinergic transients inferred from COA signals in the prefrontal cortex and hippocampus of rodents engaged in cognitive tasks (*Howe et al., 2017*; *Parikh et al., 2007*; *Teles-Grilo Ruivo et al., 2017*). Importantly, since the rate of the enzymatic reaction depends on both substrates, biosensor COA responses caused by $O_2$ transients are expected to decrease following experimental controls that have been aimed at inhibiting or removing cholinergic inputs, making this control an inadequate demonstration of the nature of COA signal (*Parikh et al., 2007*).

Our in vitro characterization of the biosensor $O_2$ dependence provided critical insights to interpret the in vivo relationship between COA and $O_2$. We found robust $O_2$-evoked phasic responses whose amplitude was anti-correlated with sensors' tonic $O_2$ dependence. Interestingly, the phasic peaks decreased with $O_2$ baseline, but were detected even under relatively high $O_2$ levels, suggesting a high likelihood of such responses to occur in vivo. Thus, our data emphasize the impact of $O_2$-evoked non-steady-state biosensor dynamics on fast time-scale in vivo measurements. This has not been described in previous studies partly because the experimental procedures used to study $O_2$-dependence were unable to unmix tonic and phasic components of sensor response (*Baker et al., 2017*; *Burmeister et al., 2003*; *Dixon et al., 2002*). Instead of generating a continuous $O_2$ increase, we induced step increments in $O_2$, allowing temporal deconvolution of sensor response components and unbiased estimation of the corresponding $K_{0.5}O_2^{app}$ values.

Remarkably, most of our observations related to sensor $O_2$-dependence were predicted by mathematical simulation of biosensor responses in vitro. The model incorporated the kinetics of the enzyme-catalyzed reaction, including enzyme-bound intermediate species, and simulated substrates' diffusion into the coating and interaction with the enzyme leading to $H_2O_2$ formation. In agreement with the experimental data, the model predicted that tonic and phasic components of $O_2$ dependence are mutually exclusive. Furthermore, our simulations suggest that increasing the enzyme loading, either by changing enzyme concentration or the coating thickness, amplifies sensor phasic responses and reduces tonic $KmO_2^{app}$. It is therefore apparent that there is no perfect combination of these two parameters that can concomitantly minimize the two components of biosensor $O_2$ dependence. In addition to reinforcing our experimental conclusions, the biosensor model provided important insights into the factors that determine phasic and tonic components of $O_2$ dependence. Simulations showed that tonic responses to $O_2$ are largely shaped by Ch depletion in the coating, which limits the linearity of the sensor response. On the other hand, phasic responses depended on the instantaneous balance between the concentration of substrates and reduced enzyme-bound intermediates, which lead to the formation of $H_2O_2$. Simulated transients were highest when the concentration of reduced intermediates and $O_2$ were in relative excess in comparison to Ch. Since variations in enzyme concentrations, coating thickness and diffusional components have mainly a quantitative effect in our estimates, these conclusions can be extrapolated to sensor geometries and enzyme coating compositions that we have not covered. Furthermore, the mathematical model of the biosensor described here provides a rigorous approach for exploration and optimization of the design of any future enzymatic biosensors and investigation of their behavior under non-stationary in vivo conditions.

The phasic $O_2$-evoked COA signals described in vitro and predicted by mathematical modeling provide crucial information to interpret the time-scale dependence of in vivo sensor dynamics following either spontaneous or exogenously evoked $O_2$ peaks. In light of those results, the temporal advancement and amplitude drop of biosensor transients relative to $O_2$, as $O_2$ rises for longer periods, is compatible with a major contribution of the phasic component of biosensor's $O_2$ dependence. This observation suggests that, in vivo, our biosensors operated in a regime close to saturation of the tonic response and highlight the effect of non-steady-state ChOx biosensor responses to $O_2$, which can be erroneously attributed to Ch.

Our observations disfavor the quantitative optimization of coating properties as a strategy to reduce sensors' $O_2$ dependence. Instead, we anticipate that strategies that increase $O_2$ accumulation in the enzyme coating (*Njagi et al., 2008*) might have relative success, although the high $O_2$ levels required to cancel phasic responses are hard to reach passively. Alternatively, some oxidase-based biosensors for in vitro applications have incorporated an electrochemical actuator that

enables local manipulation of $O_2$ concentration based on water electrolysis (*Park et al., 2006*). However, applying such design in vivo would require miniaturization and separation between the $O_2$ generation compartment and the brain to avoid electrolytic tissue damage. Furthermore, in addition to the main $O_2$ confound, one cannot completely rule out a possible modulation of COA in vivo by factors that affect enzyme conformation, including temperature and pH (*Hekmat et al., 2008*). Although the enzyme is poorly sensitive to the modest variations of these factors in vivo (*Csernai et al., 2019*; *Venton et al., 2003*), it would be relevant to characterize their potential contribution to COA dynamics in future studies. A further validation of ChOx-based measurements could be achieved by confronting the dynamics of COA with that of cholinergic signals measured with other sensing approaches, under the same experimental conditions. The latter technics include optogenetically taged single unit recordings or fluorescence reporters, which have previously revealed fast cholinergic dynamics related to arousal, sensory sampling, negative reinforcements, and unexpected events (*Eggermann et al., 2014*; *Hangya et al., 2015*; *Lovett-Barron et al., 2014*; *Reimer et al., 2016*).

In summary, our results suggest that ChOx biosensor signals in vivo are composed of a mixture of $O_2$-related artifacts and true cholinergic dynamics. The weight of each factor depends on the time-scale, with slow state-related changes reflecting cholinergic dynamics with low $O_2$-related contamination and fast transients, except for a minor fraction (less than ca. 5% in our data), being caused by phasic $O_2$ fluctuations. We show that $O_2$ transients can be triggered by cognitively relevant events, such as locomotion and periods of high SWR incidence, and confound the ChOx-based measurement of cholinergic activity. Thus, our study reveals a previously ignored phasic $O_2$-dependence of ChOx-based biosensors, which is critical to control for, particularly in the case of fine time-scale measurements of ACh. Importantly, the extent of the interplay between COA and $O_2$, and its non-linear dependence on the dynamics of enzyme substrates' concentrations at multiple time-scales made extraction of authentic phasic Ch from the in vivo COA signal unfeasible.

Our conclusions, supported by a generic mathematical modeling, can be generalized to other oxidase-based biosensors that have been used to measure neurotransmitters or metabolically relevant molecules in the brain (*Chatard et al., 2018*; *Dixon et al., 2002*; *Hascup et al., 2013*; *McMahon et al., 2007*). The exact extent of phasic and tonic $O_2$ dependence would depend on the particular enzyme kinetics and on the basal extracellular concentrations of analyte relative to the magnitude of changes in the brain.

# Materials and methods

## Key resources table

| Reagent type (species) or resource | Designation | Source or reference | Identifiers | Additional information |
|---|---|---|---|---|
| Chemical compound, drug | Chitosan | Sigma-Aldrich | 448869 | |
| Chemical compound, drug | Chloroplatinic acid hydrate | Sigma-Aldrich | 520896 | |
| Chemical compound, drug | Choline chloride | Sigma-Aldrich | C7017 | |
| Chemical compound, drug | Choline oxidase from Alcaligenes sp. | Sigma-Aldrich | C5896 | |
| Chemical compound, drug | Dopamine hydrochloride | Sigma-Aldrich | H8502 | |
| Chemical compound, drug | Gold chloride solution | Sigma-Aldrich | HT1004 | |
| Chemical compound, drug | Hydrogen peroxide | Sigma-Aldrich | 216763 | |
| Chemical compound, drug | *m*-Phenylenediamine | Sigma-Aldrich | P23954 | |
| Chemical compound, drug | *p*-Benzoquinone | Sigma-Aldrich | B10358 | |
| Chemical compound, drug | Sodium-L-Ascorbate | Sigma-Aldrich | A7631 | |

## Chemicals and solutions

All chemicals were of analytical grade, purchased from Sigma-Aldrich and used as received. Solutions were prepared in ultra-pure deionized water ($\geq$18 M$\Omega$.cm) from a Milli-Q water purification system.

## Tetrode fabrication and platings

The microelectrode support material was a 17 µm diameter Platinum/Iridium (90/10) wire insulated by a polyimide coating (California Fine Wire Company). Tetrodes were fabricated using standard methods (*Gray et al., 1995*). Briefly, four wires were twisted together and heated to melt the insulation, creating a stiff bundle of twisted wires with a total diameter of approximately 60 µm. The wires' insulation at the untwisted ending of the tetrode was then removed and the tetrode was inserted in a silica tube (150 µm inner diameter), which was glued to a holder that allowed easy manipulation of the tetrode. Next, the untwisted endings of the tetrode wires were soldered to the pins of an adapter fixed to the tetrode holder, allowing connection to the potentiostat's head-stage. Finally, the twisted ending of the tetrode was cut using micro-serrated stainless-steel scissors, leaving 1–2 cm of tetrode wire protruding out of the silica tube.

Tetrode surface treatments and platings were performed with a portable potentiostat (EmStat 3, PalmSens BV), using a freshly prepared Ag/AgCl wire (125 µm diameter, WPI inc) as *pseudo*-reference electrode. Prior to platings, electrode surfaces were cleaned by swirling the tetrode tip in isopropanol followed by an electrochemical treatment in PBS. For that purpose, we applied 70 cycles of a square wave with a first step at +1.2 V for 20 s followed by a 4 s step at −0.7 V. All tetrode sites were then gold-plated in a 3.76 µM aqueous solution of tetrachloroauric acid by applying 20 cycles of a square wave that alternated between +0.6 V for 10 s and −1.0 V for 10 s. Sites' impedances were checked after gold-plating using a nanoZ impedance tester (Multichannel Systems, GmbH). The pair of sites with the highest impedance was then platinized in a 10 mM chloroplatinic acid solution in 0.1 M sulfuric acid by DC amperometry at −0.1 V, until a current of −30 nA was reached.

## TACO sensor coatings

Choline oxidase immobilization was performed as previously described (*Santos et al., 2015*). Briefly, a 0.5% (w/v) chitosan stock solution was solubilized in saline (0.9% NaCl) under stirring at pH 4–5, adjusted by addition of HCl. After solubilisation, the pH was set to 5–5.6 by stepwise addition of NaOH.

To form a chitosan/ChOx cross-linked matrix, 1.5 mg of p-benzoquinone was added to 100 µL of 0.2% chitosan. Four µL of this solution were then mixed with a 4 µL aliquot of ChOx at 50 mg/mL in saline. The tetrode tip was coated by multiple dips (10-15) in a small drop of ChOx immobilization mixture, created using a microliter syringe (Hamilton Co.). The microelectrode and syringe were micromanipulated under a stereomicroscope. The coating procedure was stopped when the chitosan/protein matrix was clearly visible under the microscope.

Following enzyme immobilization, tetrode site's response to Ch was tested and *meta*-phenylenediamine (*m*-PD) was electropolymerized on the pair of sites with the highest sensitivity (please see *Biosensor calibrations* sub-section). Electropolymerization was performed in a nitrogen bubbled oxygen-free PBS solution of 5 mM *m*-PD by DC amperometry at +0.6 V during 1500s. The biosensors were stored in air and calibrated on the day after *m*-PD electropolymerization.

## Biosensor calibrations

All in vitro tests were done in a stirred calibration buffer kept at 37°C using a circulating water pump (Gaymar heating/cooling pump, Braintree Scientific, Inc, USA) connected to the calibration beaker. Routine calibrations after enzyme immobilization and *m*-PD electropolymerization steps were performed by amperometry at a DC potential of +0.6 V vs. Ag/AgCl *pseudo*-reference electrode. After stabilization of background current in PBS, sensors were calibrated by three consecutive additions of 10 µM Ch followed by 4.9 µM $H_2O_2$. In the case of complete (*m*-PD polymerized) biosensors, the response to 1 µM DA and 100 µM AA was also tested. Voltammograms of $H_2O_2$ were done by consecutive additions of 4.9 µM $H_2O_2$ at different applied DC voltages.

In vitro $O_2$ tests were carried out in a sealed beaker. After addition of 5 µM Ch, the calibration buffer was bubbled with nitrogen during approximately 30 min. Then, known $O_2$ concentrations (5 or 10 µM) were added to the medium from an $O_2$-saturated PBS solution previously bubbled with pure $O_2$ during 20 min. Biosensor response to $O_2$ additions was measured at +0.6 V *vs.* Ag/AgCl. In a set of calibrations, $O_2$ was concurrently measured by polarizing a gold-plated channel at negative voltage, while keeping the remaining at +0.6 V. The electrode response to $O_2$ was characterized by obtaining $O_2$ voltammograms by calibrations that consisted in three additions of 5 µM $O_2$,

performed at different applied DC potentials (*Figure 3A*). In these experiments, $O_2$ was purged from the solution after each voltage step. Although the gold-plated site could also reduce $H_2O_2$, the very low response magnitude as compared to $O_2$ (*Figure 3A*) assured a negligible contribution of $H_2O_2$ generated by immobilized ChOx to the $O_2$ signal. To avoid electrical cross-talk between tetrode sites, sporadically observed when we applied a high negative voltage to one site, we set $-0.2$ V vs. Ag/AgCl as the standard voltage for $O_2$ measurements both in vitro and in vivo. At this holding potential, neither current cross-talks nor a detectable effect on local $O_2$ levels (that could affect COA signal) were observed (*Figure 3—figure supplement 1*).

Sensor response times to Ch and $O_2$ were estimated using the above-mentioned calibration setups.

## Experimental model and subject details

Freely moving recordings were performed on a 6 months old Long-Evans rat and head-fixed recordings were done in a total of six 3–7 month old C57BL/6J mice. Variability from posthoc characterization of the sensor performance, as well as from inferred electrode localization made sample size predetermination unreliable. We set a minimal biological sample size of 3 for any analyzed parameter, which was found to be appropriate given the consistency of the in vivo data. Additionally, we performed a large sample study in vitro. All experimental procedures were established, and have been approved in accordance with the stipulations of the German animal welfare law (Tierschutzgesetz)(ROB-55.2–2532.Vet_02-16-170).

## Surgeries

For freely moving recordings, we chronically implanted a tetrode biosensor and a 32-channel linear silicon probe array (A1 × 32-7mm-100–1250 H32, NeuroNexus Technologies, Inc) in the rat brain. The general procedures for chronic implantations of electrode arrays have been described in detail (*Vandecasteele et al., 2012*). Prior to surgery, the tetrode biosensor and the silicon probe array were attached to home-made microdrives. Silicon probe's sites were then gold-plated until impedances at 1 kHz decreased below 200 kΩ (*Ferguson et al., 2009*). Anesthesia was induced with a mixture of Fentanyl 0.005 mg/kg, Midazolam 2 mg/kg and Medetomidine 0.15 mg/kg (MMF), administered intramuscularly. The rat was continuously monitored for the depth of anesthesia (MouseStat, Kent Scientific Corporation, Inc). After the MMF effect washed out, anesthesia was maintained with 0.5–2% isoflurane via a mask, and metamizol was then subcutaneously administered (110 mg/kg) for analgesia. The tetrode biosensor was implanted in the cortex above the right dorsal hippocampus (AP $-3.7$ mm, ML $-2.5$ mm, DV $-1.2$ mm, relative to bregma) and the silicon probe array was implanted at 0.8 mm posterior from it, spanning most cortical and hippocampal layers (AP $-4.5$, ML $-2.4$, DV $-3.4$). The microdrives were secured to the skull with a prosthetic resin (Paladur, Kulzer GmbH). An Ag/AgCl (125 μm thick) silver wire coated with Nafion (*Hashemi et al., 2011*) was inserted in the cerebellum and served as the *pseudo*-reference electrode for electrochemical recordings. The ground for electrophysiology was a stainless-steel screw implanted at the surface of the cerebellum. To reduce line noise, this Ag/AgCl wire was shorted with the electrophysiology ground at the input of the electrochemical head-stage.

Mice used in head-fixed recordings were implanted with a head-post. Anesthesia followed the same procedures as in rats. A mixture of 0.05 mg/kg Fentanyl, 5 mg/kg Midazolam, and 0.5 mg/kg Medetomidine was administered intraperitoneally to induce anesthesia, which was later maintained with isoflurane and Metamizol (200 mg/kg). A craniotomy was made above the dorsal hippocampus and a Nafion-coated Ag/AgCl wire was implanted in the cerebellum. Depending on the head-post configuration, it was cemented either to the back of the skull above the cerebellum or above the hemisphere contralateral to the craniotomy, using UV-curing dental cement (Tetric EvoFlow, Ivoclar Vivadent AG). Finally, the craniotomy and surrounding skull were covered with a silicone elastomer (KWIK-CAST, World Precision Instruments Inc).

## Electrochemical and electrophysiological equipment and recordings

Amperometric measurements were performed using either a four-channel (MHS-BR4-VA) or a 8-channel (MBR08-VA) potentiostat connected to four- or eight-channel miniature head-stages, respectively (npi electronic GmbH, Germany). In addition to providing a higher channel count, the

MBR08-VA allowed independent control of the potential applied to each channel. This feature enabled simultaneous measurement of the biosensor signal, arising from ChOx, and $O_2$. The DC analog signal from the head-stage was amplified and digitized at 30 kHz and stored for offline processing using the Open Ephys acquisition board and GUI (*Siegle et al., 2017*).

Freely moving electrochemical recordings were done using the MHS-BR4-VA potentiostat and the corresponding four-channel miniature head-stage. Electrophysiological signals were pre-amplified using a 32-channel head-stage with 20x gain (HST/32 V-G20, Plexon Inc) which was connected to a multichannel acquisition system (Neuralynx Inc). Data was acquired at 32 kHz and stored for offline processing. Both head-stages were connected to the respective recording systems via light and flexible cables suspended on a pulley so as not to add weight to the animal's head. The tetrode biosensor was gradually lowered through the cortex until it reached the hippocampal CA1 pyramidal layer. Correct targeting was assessed based on brain atlas coordinates and by the identification of hippocampal ripples. Recordings were performed in a square open-field arena (1.5 m x 1.5 m), where the animal could sleep or explore the environment at will. Chocolate sprinkles were occasionally spread on the maze to enforce exploratory behavior. The position of the rat head was derived from small reflective markers attached to the chronic implant. A motion capture system consisting of multiple infrared cameras (Optitrack, NaturalPoint Inc) was used to 3D-track the markers with high spatio-temporal resolution (data acquired at 120 Hz).

Head-fixed recordings in mice were performed using the MBR08-VA potentiostat and respective head-stage. After fixing the mouse, the layer of silicone elastomer protecting the craniotomy was removed. The dura matter above the target brain region was removed and the tetrode biosensor was slowly inserted through the cortex until the hippocampal CA1 pyramidal layer was reached. Accurate targeting was assessed according to brain atlas coordinates and/or by the online identification of hippocampal ripples in the recording. In 5 out of 10 recording sessions mice were head-fixed on a cylindrical treadmill. Movement was quantified based on the video optical flow arising from treadmill rotation using Bonsai (*Lopes et al., 2015*). In the remaining sessions, mice were head-fixed on a rotating disc which encoded its turns. The analog signal from the disc encoder was fed into the Open Ephys acquisition board and used to quantify mice locomotion.

## Data analysis

Raw recordings were preprocessed by low-pass filtering and resampling at 1 kHz. All data analysis was done in Matlab using custom-made functions (MathWorks).

### In vitro sensor responses

In vitro analysis of biosensor responses was performed on 10 Hz downsampled data, low-pass filtered at 1 Hz. Sensitivities to Ch and $O_2$ were determined by linear-regression of the responses to the first three analyte additions, whereas the sensitivities to $H_2O_2$ and interferents were estimated from a single addition. Following *pseudo*-sentinel subtraction (Au/Pt/*m*-PD - Au/*m*-PD sites), selectivity ratios for the COA measurement were estimated by dividing the mean of Ch sensitivity by the mean of interferent responses. The biosensors' limit of detection (LOD) for Ch, extracted from the COA signal, was calculated as the Ch concentration corresponding to three times the baseline standard deviation (SD). The $T_{50}$ and $T_{90}$ response times were defined as the time between the onset of current increase in response to analyte and 50% or 90% of the maximum current, respectively.

### Artifact cancellation by common-mode rejection

In vivo electrochemical signals from sites sensitive to COA were cleaned by subtraction of the respective 1 kHz data by the corresponding *pseudo*-sentinel channel upon a frequency-domain correction. The latter procedure has been described in detail and optimizes common-mode rejection by correcting phase and amplitude mismatches between channels arising from slight frequency-dependent variations in impedance (*Santos et al., 2015*), a procedure conceptually analogous to orthogonalization of the EEG signals (*Hipp et al., 2012*). This correction was based on the estimation of a transfer coefficient (*T*) describing the transfer function from the currents derived from the pseudo-sentinel to the COA-sensitive channel in the frequency domain, according to the following equation:

$$COA_{mPD} = iFFT(Au/Pt/mPD(jw) - T(jw)Au/mPD(jw)), \quad (1)$$

where $jw$ is a complex value at frequency $w$ and $COA_{mPD}$ is the clean COA signal obtained from sites with $m$-PD (note that $COA_{mPD}$ was used as the measure of COA dynamics throughout this study).

Upon applying a fast Fourier transformation (FFT), each signal can be described by its amplitude and a phase across a range of frequencies. The amplitude of $T$ for each FFT frequency bin was then estimated from the square root of ratio between the power of the platinized channel carrying the COA signal and the pseudo-sentinel channel. The phase of T was estimated from the phase shift of the cross-spectrum, reflecting the difference between the phases of the two signals at each frequency, during time-windows with high phase-locking ($>0.9$) in each frequency bin (corresponding to periods of high consistency of phase-shift values). These estimates were computed for each biosensor used in vivo from average spectra obtained from multiple slow-wave periods devoid of movement artifacts. In the low-frequency range ($<0.3$ Hz), due to COA contribution to power in the platinized channel, the estimation of the amplitude of $T$ was done by linear extrapolation considering the trend at contiguous higher frequencies. The cleaned COA signal was obtained by inverse FFT of the corrected subtraction in the frequency domain. Cleaned $O_2$ signals were obtained following the same logic:

$$O_2 = iFFT(Au_{-0.2V}(jw) - T(jw)Au/mPD(jw)), \qquad (2)$$

where $Au_{-0.2V}$ represents the $O_2$ measuring channel polarized at $-0.2$ V vs. Ag/AgCl whereas the Au/mPD channel (at $+0.6$ V vs. Ag/AgCl) was used as the *pseudo*-sentinel. To substantiate the validation of the in vivo COA measurement, two additional cleaned signals were computed:

$$COA_{non-mPD} = iFFT(Au/Pt(jw) - T(jw)Au(jw)), \qquad (3)$$

$$NCC = iFFT(Au(jw) - T(jw)Au/mPD(jw)), \qquad (4)$$

The $COA_{non-mPD}$ signal results from the pair of sites lacking $m$-PD, whereas the neurochemical confounds signal (NCC) represents the dynamics of the mixed contribution of neurochemical confounds (e.g. ascorbate, dopamine) picked-up by the $Au$ site.

The cleaned signals (*Equations 1-4*) were then low-pass filtered at 1 Hz and downsampled to 10 Hz for most of the analysis excluding time lags, which were computed on 100 Hz downsampled data.

## Brain state separation

Local-field potential-related power spectrograms were computed using custom-made Matlab functions based on multi-taper analysis methods (*Mitra and Pesaran, 1999*). Separation of brain states in freely moving recordings was based on LFP spectral features and behavior. Active wake states were defined as periods when the animal moved vigorously and continuously ($>30$ s) and showed a prominent LFP spectral peak in theta range (6–10 Hz). Quiet wakefulness or immobility was defined as a period without prominent theta and with occasional movements ($<30$ s between movement bouts). Long periods ($>1$ min) without movement and without prominent theta, rather showing high delta power (1–4 Hz) were ascribed to NREM sleep. Rapid eye movement sleep was detected as periods showing a sustained theta band ($>30$ s) and negligible movement.

## Sharp-wave/ripples and related biosensor signals

To detect SWRs, the wide-band electrochemical or electrophysiological signal was band-pass filtered (120–200 Hz), squared and smoothed with a 4.2 ms standard deviation and 42 ms wide Gaussian kernel. The square root of this trace was then used as the power envelope to detect oscillatory bursts. The events exceeding the 98th percentile of the power envelope distribution, having at least five cycles and lasting less than 200 ms were detected as ripples. For the analysis of correlations between integrated ripple power and SWR-triggered sensor signals, a ripple power envelope was obtained upon Hilbert-transforming the band-pass filtered electrochemical signal (ripple band, 120–200 Hz). Different ripple integration times were obtained by smoothing the power envelope with moving average windows of different lengths. Correlations were then computed between smoothed ripple power envelopes at SWR times and the corresponding changes in COA or $O_2$ (relative to their baseline value 1 s prior to SWRs) at different lags from SWRs.

The amplitudes of SWR-related COA and $O_2$ were obtained from the difference between the values at SWR lags corresponding to peaks and onsets, extracted from average SWR-triggered traces.

## Locomotion bouts

To detect locomotion bouts in freely moving recordings, speed was computed from the derivative of low-pass filtered position (0.5 Hz). Speed was then band-pass filtered (0.02–0.2 Hz) and locomotion bouts were detected as peaks in speed that exceeded a manually set threshold. In head-fixed recordings on the disc, mouse locomotion was derived from its rotation in 1 s bins. When head-fixed on the treadmill, mouse locomotion was quantified based on the optic flow from a recorded video, choosing a region of interest that covered only a moving part of the treadmill. The signal was resampled to 1 Hz in order to match the sampling rate of locomotion on the rotating disc. Locomotion bouts were detected as peaks on the band-pass filtered speed (0.02–0.2 Hz) that exceeded a manually defined threshold.

The amplitude of locomotion-bout-associated COA and $O_2$ signal transients was calculated based on the difference between the values at manually determined times of transient onsets and peaks, associated with each event. Likewise, the times of locomotion bout onsets (used to calculate speed change) and the associated peaks in theta power were manually defined based on visual inspection of speed time courses and LFP spectrograms, respectively.

## Oxygen-related ChOx transient signals analysis

The amplitude of broad-band spontaneous and exogenously induced $O_2$ transients and COA transients associated with them was calculated, for each event, as the difference between the peak value and that at semi-automatically defined time of the transient's onset. For the detection of spontaneous $O_2$ transients occurring outside periods with SWRs and locomotion, the $O_2$ peaks occurring within −5 s to +1 s from SWRs or within −14 s to + 4 s from peaks in speed were excluded.

To capture and separately analyze spontaneous COA and $O_2$ transients of highly variable timescale (1–14 s) and non-harmonic shape we performed peak detection on the two signals band-pass filtered in a frequency range [*f/2 f*] with corner frequency *f*, defining each filter, varying from 0.05 to 0.5 Hz. Peaks of the transients were then detected in both band-passed COA and $O_2$ signals, selecting the events that exceeded half of the maximal peak amplitude. In each frequency band, COA transients used for the analysis were restricted to be within *0.75/f* of any $O_2$ peak. The amplitudes of $O_2$ and COA transients were defined as peak magnitudes derived from the band-pass filtered signals. For each filter band, the linear relationship between COA and $O_2$ peak amplitudes was estimated as a slope of the linear model fit to all detected event pairs for this filter. To interpret the nonlinear shape of the $O_2$ transients captured by each filter, we extracted rise time (from trough to peak) from the median peak aligned $O_2$ transients and used these values in *Figure 7E–H* and *Figure 7—figure supplement 1*.

The transient onsets and peaks of exogenously induced changes in $O_2$ and COA were manually detected.

## Modeling biosensor responses in vitro

We simulated biosensor responses in vitro by numerically solving a system of partial differential equations that describe the diffusion of the substrates Ch and $O_2$ in the enzyme coating and their interaction with the enzyme, leading to product formation.

The buffer solution where the biosensor was placed for calibration is a free-flow environment, in which the concentrations of Ch and $O_2$ are constant over time. Therefore, considering R is the coating thickness, at the boundary between the biosensor coating and the calibration buffer Ch is kept constant at 5 µM during the calibration.

$$Ch(R,t) = 5 \text{ and } \left.\frac{\partial Ch}{\partial t}\right|_R = 0 \tag{5}$$

Oxygen is changed in steps (*x*), starting from 0, during the calibration, so between $O_2$ step increases we have:

$$O_2(R,t) = x \text{ and } \left.\frac{\partial O_2}{\partial t}\right|_R = 0 \tag{6}$$

Enzyme substrates (Ch and $O_2$) diffuse from the bulk solution into the enzyme layer, eventually reaching the electrode site, whereas $H_2O_2$ is locally generated and diffuses within the coating. As the size of our recording sites is very small, this process is better described by a spherical diffusion equation:

$$Difs = \frac{\partial[S]}{\partial t} = D_s \frac{1}{r^2}\frac{\partial}{\partial r}\left(r^2\frac{\partial[S]}{\partial r}\right) \tag{7}$$

where $S$ represents substrates or $H_2O_2$ concentration, $D_S$ is the respective diffusion coefficient, and $r$ is the distance to the electrode surface.

In order to simulate realistic two-substrate biosensor responses, we modeled the formation of enzyme intermediate complexes resulting from Ch binding to the enzyme and $O_2$ oxidation reactions, which have been described in detail (*Fan and Gadda, 2005*; *Figure 4—figure supplement 2*). Briefly, enzyme-bound Ch (*ECh*), which is in equilibrium with the free reactant species (*E* and *Ch*), undergoes a chemical step leading to the reduction of the FAD enzyme prosthetic group. In this nearly irreversible step, Ch is converted to betaine aldehyde, which remains mostly enzyme-bound ($E_{red}BA$). The first step in which $H_2O_2$ is produced results from the oxidation of $FAD_{red}$ by $O_2$ ($E_{ox}BA$) followed by a second chemical step in which $FAD_{ox}$ is reduced by betaine aldehyde. The resulting enzyme-bound glycine betaine ($E_{red}GB$) is then oxidized by $O_2$, producing $H_2O_2$. The reaction cycle is completed with release of glycine betaine bound to FAD-oxidized enzyme ($E_{ox}GB$).

The instantaneous change in the concentration of enzyme substrates, free enzyme, enzyme-bound intermediate complexes and reaction products can then be described by the following system of partial differential equations:

$$
\begin{aligned}
\frac{\partial E}{\partial t} &= -kf[E][Ch] + kr[ECh] + k_5[E_{ox}GB] \\
\frac{\partial[Ch]}{\partial t} &= Dif_{Ch} - kf[E][Ch] + kr[ECh] \\
\frac{\partial[ECh]}{\partial t} &= kf[E][Ch] - kr[ECh] - k_1[ECh] \\
\frac{\partial[O_2]}{\partial t} &= DifO_2 - k_2[E_{red}BA][O_2] - k_4[E_{red}GB][O_2] \\
\frac{\partial[E_{red}BA]}{\partial t} &= k_1[ECh] - k_2[E_{red}BA][O_2] \\
\frac{\partial[E_{ox}BA]}{\partial t} &= k_2[E_{red}BA][O_2] - k_3[E_{ox}BA] \\
\frac{\partial[E_{red}GB]}{\partial t} &= k_3[E_{ox}BA] - k_4[E_{red}GB][O_2] \\
\frac{\partial[E_{ox}GB]}{\partial t} &= -k_5[E_{ox}GB] + k_4[E_{red}GB][O_2] \\
\frac{\partial[H_2O_2]}{\partial t} &= Dif_{H_2O_2} + k_2[E_{red}BA][O_2] + k_4[E_{red}GB][O_2]
\end{aligned}
\tag{8}
$$

Note that we ignored the kinetics of glycine betaine formation, as it is not relevant regarding the sensor signal transduction and it would not affect the concentrations of any reaction species.

In the beginning of the simulated calibration, there is no Ch in the coating, so all enzyme molecules are free, at a concentration that equals the total enzyme concentration ($[E]_0$).

$$[E](r,t_0) = [E]_0 \quad r = r_1,...,R. \tag{9}$$

In turn, in the same conditions, there are no enzyme-bound species and no product formed.

$$
\begin{aligned}
[ECh](r,t_0) = [E_{red}BA](r,t_0) = [E_{ox}BA](r,t_0) = [E_{red}GB](r,t_0) \\
= [E_{ox}GB](r,t_0) = [H_2O_2](r,t_0) = 0, \quad r = r_1,...,R.
\end{aligned}
\tag{10}
$$

At the electrode surface, there is zero flux of substrates and $H_2O_2$ is rapidly oxidized, whereas at the boundary between the coating and bulk solution it is rapidly washed away.

$$\left.\frac{\partial Ch}{\partial t}\right|_{r_1} = 0, \left.\frac{\partial O_2}{\partial t}\right|_{r_1} = 0, \ [H_2O_2](r_1) = 0 \text{ and } [H_2O_2](R) = 0. \tag{11}$$

The sensor response current is then given by the charge of two-electrons per each oxidized $H_2O_2$ molecule times the flux of $H_2O_2$ at the electrode surface.

$$I = 2F(\pi r_e^2)J_{H_2O_2}, \ J_{H_2O_2} = D_{H_2O_2}\frac{\partial H_2O_2}{\partial r}|_{r_1}, \tag{12}$$

where $F$ is the Faraday's constant and $\pi r_e^2$ is the electrode surface area.

Considering the initial conditions described above the system of partial differential equations was numerically solved by discretization in space and time (time and space steps $dt$ = 0.1 ms and $dr$ = 1 μm, respectively) using the finite difference approximation method (*Baronas et al., 2009*).

The values ascribed to the variables used in the model are summarized in *Table 2*. The rate constants corresponding to the reaction mechanism in *Figure 4—figure supplement 2* were extracted from the corresponding literature (*Fan and Gadda, 2005*). An enzyme concentration in the coating of 263 uM was estimated from its concentration in the mixture used for coatings, ignoring drying effects upon coating. Variations in enzyme concentration were simulated around that value. We estimated the diffusion coefficients of enzyme substrates and $H_2O_2$ in the coating taking into account the free diffusion coefficients in solution multiplied by a hindrance factor α. The latter was set at 0.8, considering the expected effect of macromolecular crowding on diffusion, for protein concentrations in the range of those used in our simulations (*Lamers-Lemmers et al., 2000*; *Santos et al., 2011*).

## Statistical analysis

All statistical tests were performed using Matlab. Prior to statistical comparisons, the normality of the data was tested by a Anderson-Darling test. In the case of non-normal distributions, comparisons between two groups were performed using a non-parametric two-sided sign test (signtest, Matlab). To test the effect of two factors on non-normal data (effect of ripple count and power on COA or $O_2$ transients), an aligned-rank-transformation (*Wobbrock et al., 2011*) was applied followed by two-way ANOVA for unbalanced data (nanova, Matlab). Whenever normality could not be discarded, two-sided two-sample *t*-tests (paired or unpaired) were used to compare two groups. Two-sided one-sample *t*-tests were used to test deviations from the null hypothesis (zero). Multiple comparisons were done by one-way ANOVA for unbalanced data. The effect of two or more factors was accounted for by two- or three-way ANOVA for unbalanced data.

Data were presented as average ±95% confidence interval (CI) to allow easy assessment of the significance of estimates. Average corresponds to the mean in normal data or to the median otherwise. The CI of medians was computed based on fractional order statistics (*Hutson, 1999*). Confidence intervals for correlation coefficients and COA vs. $O_2$ slopes were computed from the percentiles 2.5 and 97.5 of bootstrapped data.

**Table 2.** Values of constants used in the biosensor model.

| Constant | Value | Reference |
|---|---|---|
| $k_f^*$ | $2 \times 10^6 \ M^{-1} \ s^{-1}$ | *Fan and Gadda, 2005* |
| $k_r^*$ | $580 \ s^{-1}$ | *Fan and Gadda, 2005* |
| $k_1$ | $93 \ s^{-1}$ | *Fan and Gadda, 2005* |
| $k_2$ | $8.64 \times 10^4 \ M^{-1} \ s^{-1}$ | *Fan and Gadda, 2005* |
| $k_3$ | $135 \ s^{-1}$ | *Fan and Gadda, 2005* |
| $k_4$ | $5.34 \times 10^4 \ M^{-1} \ s^{-1}$ | *Fan and Gadda, 2005* |
| $k_5$ | $200 \ s^{-1}$ | *Fan and Gadda, 2005* |
| $D_{Ch}^\dagger$ | $1197 \ \mu m^2 \ s^{-1}$ | *Valencia and González, 2012* |
| $D_{O2}^\ddagger$ | $2500 \ \mu m^2 \ s^{-1}$ | *Santos et al., 2011* |
| $D_{H2O2}$ | $1830 \ \mu m^2 \ s^{-1}$ | *van Stroe-Biezen et al., 1993* |
| α | 0.8 | |

* Estimated based on $K_d$, $K_{cat}$ and $K_m$ values of ChOx for choline.

†Estimated using the Stokes−Einstein Gierer-Wirtz Estimation method.

‡Extrapolated from room temperature to 37˚C using a factor of 2.6% per degree (*Han and Bartels, 1996*).

## Acknowledgements

We thank Francisco Almeida and Kenneth Klau for technical assistance.

## Additional information

### Funding

| Funder | Grant reference number | Author |
|---|---|---|
| Bundesministerium für Bildung und Forschung | 01GQ0440 | Anton Sirota |
| Deutsche Forschungsgemeinschaft | Munich Cluster for Systems Neurology (SyNergy EXC 1010) | Anton Sirota |
| European Union Framework Programme for Research and Innovation | FETPROACT program via grant agreement no.723032 (BrainCom) | Anton Sirota |
| Deutsche Forschungsgemeinschaft | Priority Programs 1665 and 1392 | Anton Sirota |

The funders had no role in study design, data collection and interpretation, or the decision to submit the work for publication.

### Author contributions

Ricardo M Santos, Conceptualization, Data curation, Software, Formal analysis, Investigation, Visualization, Methodology, Writing - original draft, Writing - review and editing; Anton Sirota, Conceptualization, Supervision, Funding acquisition, Project administration, Writing - review and editing

### Author ORCIDs

Ricardo M Santos (iD) https://orcid.org/0000-0001-6182-1708
Anton Sirota (iD) https://orcid.org/0000-0002-4700-6587

### Ethics

Animal experimentation: All experimental procedures were established, and have been approved in accordance with the stipulations of the German animal welfare law (Tierschutzgesetz ) (ROB-55.2-2532.Vet_02-16-170).

### Decision letter and Author response

Decision letter https://doi.org/10.7554/eLife.61940.sa1
Author response https://doi.org/10.7554/eLife.61940.sa2

## Additional files

### Supplementary files

• Transparent reporting form

### Data availability

Source data and code for figures 1,2, 5-8 as well as raw and intermediate data for the analysis in Figure 3 are provided. The code used to model biosensor responses and obtain the plots in Figure 4 is also available. All data are deposited at https://doi.org/10.5281/zenodo.4564638.

The following dataset was generated:

| Author(s) | Year | Dataset title | Dataset URL | Database and Identifier |
|---|---|---|---|---|
| Sirota A, Santos RM | 2021 | Phasic oxygen dynamics confounds fast choline-sensitive biosensor | https://doi.org/10.5281/zenodo.4564638 | Zenodo, 10.5281/zenodo.4564638 |

signals in the brain of behaving rodents

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
