## [Decision Letter]

**Acceptance summary:**

As one of the major neuromodulators, acetylcholine signaling plays vital roles in shaping cortical functions. Therefore, faithfully monitoring and measuring acetylcholine dynamics in vivo under behavioral relevant context is important for our understanding of cholinergic modulation. In this study, Santos and Sirota built upon previously used choline oxidase-based electrochemical biosensors and developed a novel acetylcholine sensor with much improved properties, including simultaneous measurement of choline oxidase activity and oxygen levels within the same tetrode. Using the new sensor in freely-moving and head-fixed rodents, the authors observed strong correlation between choline oxidase and oxygen in vivo. The authors then obtained experimental evidence both in vitro and in vivo in combination with mathematical modeling to suggest that measurement using electrochemical biosensors based on choline oxidase is confounded by oxygen dynamics itself. These findings challenge the belief that choline oxidase-based sensors can measure sub-second temporal dynamics of choline concentrations in vivo, and also calls for critical re-evaluation of all oxidase-based biosensors literature to determine the contribution of phasic oxygen dynamics in vivo.

**Decision letter after peer review:**

Thank you for submitting your article "Phasic oxygen dynamics underlies fast choline-sensitive biosensor signals in the brain of behaving rodents" for consideration by *eLife*. Your article has been reviewed by Joshua Gold as the Senior Editor, a Reviewing Editor, and three reviewers. The following individuals involved in review of your submission have agreed to reveal their identity: George Wilson (Reviewer #2); Shih-Chieh Lin (Reviewer #3).

The reviewers have discussed the reviews with one another and the Reviewing Editor has drafted this decision to help you prepare a revised submission.

Summary:

In the manuscript, the authors developed a novel tetrode-based amperometric choline oxidase (ChOx) sensor that can simultaneously measure ChOx and O_2_ levels within the same tetrode. This sensor allowed the authors to observe strong correlations between ChOx and O_2_ levels in vivo in behaving rats and mice and under several distinct behavioral contexts. The authors further combined in vivo as well as in vitro perturbation experiments to demonstrate that phasic fluctuations in O_2_ concentration can lead to fluctuations in ChOx measurements. These findings also challenge the long-held belief that ChOx sensors can measure sub-second temporal dynamics of choline concentrations in vivo, and also calls for critical re-evaluation of all oxidase-based biosensors literature to determine the contribution of phasic O_2_ dynamics in vivo.

Essential revisions:

The following two main concerns need to be addressed:

1) Further characterization and clarification are needed regarding the sensor properties. This is crucial for the potential users in the field to judge and use the sensor, and for interpretation of the biology results using the sensor.

i) Clear statement in prominent places about the improvement of the sensor and new potential for its biologic applications separating from the authors' 2015 paper.

ii) Regarding the choline responses: characterizing the linearity of choline response is important for users to understand the sensor properties. Related, demonstration how to calibrate moving artificial signals in freely-moving rodents will be useful for the future applications. Further, since the COA signal is confounded by phasic O_2_ fluctuations, the authentic changes in COA are potentially interfered by O_2_-evoked enzymatic responses. The interpretation of the signal interference needs to be clearly discussed, including O_2_-evoked changes, and other related signaling changes, like DA.

iii) The dimensions of the sensor head need to be specified and spelled out clearly. It seems to be around 50 um, but the text seems to suggest 150 um. The individual sensing elements are 17 μm in diameter. If this is true, it is very exciting because it exhibits hemispherical diffusion yielding higher response and enhanced sensitivity. This may improve spatial and temporal resolution if this is in indeed a much smaller sensor as a disk-shaped one.

iv) The role of the sentinels with differential plating is very interesting, but the function of the sentinels is not clear (Introduction "canceling LFP-related currents"). They consume oxygen. Why does this not result in overlap of the diffusion layer for the choline sensor and therefore affect choline response? Please explain why differential electroplating was employed.

v) Please explain how time-dependent behavior of the sensor was measured. This process typically leads to the formation of a film on this electrode surface which can affect sensitivity. According to the authors' 2015 paper, the method for measuring the response time seems rather crude, and may overestimate the response time which is related to the mixing of the solution. This needs to be discussed.

vi) The effect of LFP and other perturbations of sensor responses need to be more clearly explained.

2) Re-organization of the manuscript to improve the readability. This manuscript contains the characterization of the TACO sensor and application of this sensor to monitor real-time behavior in freely moving rodents. The design and characterization of the sensor is intermingled with the application of studying the choline biology with the sensor, making the logic flow hard to follow. The arrangement and presentation of the figures need to be improved so readers can appreciate both characterization and applications aspects and how they are tightly linked. This might also involves properly arrange main figures and associated supplementary figures.

Reviewer #1:

Santos and Sirota developed a novel Tetrode-based Amperometric ChOx (TACO) sensor. This multichannel configuration can simultaneously measure the ChOX activity (COA) and O_2_ in the same brain spot. Using the TACO sensor in freely-moving and head-fixed rodents, they found that COA and O_2_ dynamics following locomotion in active state and hippocampal sharp-wave/ripple (SWR) complexes during quiescence state. It's interesting that the COA signal can be calibrated by subtraction of the pseudo-sentinel from the Ch-sensing sites signal the TACO sensor. However, the COA signal is confounded by phasic O_2_ fluctuations, so, the authentic changes in COA are interfered by O_2_-evoked enzymatic responses. This question isn't addressed in this paper.

1) The author found that the COA readout is confounded by phasic O_2_ fluctuations in in vitro and in vivo experiments. These results cast doubt on the validity of the authentic cholinergic response in freely-moving or head-fixed rodents. These findings seem to be generalized to other oxidase-based biosensors, although the author has some discussion on how to address this question. However, we can't get authentic cholinergic dynamic in vivo by TACO biosensor if we didn't clear the biosensor O_2_ dependence. So, the author should try to address this question.

2) The author should demonstrate that how to calibrate moving artificial signals in freely-moving rodents.

3) Concerns on the selectivity. Figure 1E shows the TACO sensor also responses to dopamine and ascorbate. The author should demonstrate the selectivity of TACO sensor on different monoamines at different concentrations.

Reviewer #2:

This is an important piece of work addressing in-vivo measurements where two coupled components are to be measured ideally in the same time and space components. Unfortunately, the impact of this work is likely to be minimized due to its poor organization and the attempt to deal with a number of separate but related issues in the same manuscript. Accordingly, it is suggested that this work be divided into two manuscripts to be published together. The focus of the two might be:

A) A Tetrode-Based Microsensor (TACO) – This work would focus on the criteria for performance that would be expected for a new sensor, presumably with new and unique properties. This work would include differential plating of electrodes (It is unclear whether some or all of this work has been previously reported), dimension considerations, and the simulation of sensor response. (An important consideration but frequently overlooked is that a sensing element with a 17um diameter will exhibit hemispherical diffusion (Equation 4)). Other issues such as interferences, stability, sensor response time and linearity need to be more fully explained. Presumably such a sensor configuration would be useful in other applications involving oxidase-based sensors.

B) Effects of Local Field Potential and Oxygen-evoked ChOx transients in the In-Vivo Measurement of Acetylcholine in Freely Moving Rodents (A better title can surely be presented!) – Here the focus should be on the in-vivo measurements including a qualitative explanation of the LFP and O_2_ response and how the TACO sensor corrects for this. Presumably the detection of REM and NREM sleep will be detected by EEG. This is not well explained. Also unclear is how the improved performance of the sensor affects the conclusions of the in-vivo studies.

Reviewer #3:

In this manuscript, Santos and Sirota demonstrated that the in vivo fast choline dynamics detected using choline-oxidase based biosensors is strongly correlated with, and likely caused by, phasic oxygen dynamics in vivo. The authors developed a novel tetrode-based amperometric choline oxidase (ChOx) sensor that can simultaneously measure ChOx and O_2_ levels within the same tetrode, which enabled the authors to observe strong correlations between ChOx and O_2_ levels in vivo (in behaving rats and mice, and under several distinct behavioral contexts). To dissect the causal relationship and determine the role of phasic O_2_ transients, the authors further combined in vivo as well as in vitro perturbation experiments to demonstrate that phasic fluctuations in O_2_ concentration can lead to fluctuations in ChOx measurements. Moreover, mathematical modeling recapitulates the systemic relationship between ChOx and O_2_, suggesting the source of this coupling stems from non-steady-state enzyme kinetics. Together, these findings challenge the long-held belief that ChOx sensors can measure sub-second temporal dynamics of choline concentrations in vivo, and also calls for critical re-evaluation of all oxidase-based biosensors literature to determine the contribution of phasic O_2_ dynamics in vivo.

The study provides extensive evidence to support their claim: correlational, causal, analytical and modeling. The authors employed multiple levels of approaches, from the development of novel biosensors that leads to the observed correlation, to careful in vivo and in vitro perturbation experiments to demonstrate causal relationship. The data is carefully analyzed, and elegantly matched with modeling results. The results of this study have broad implications beyond the ChOx literature and in fact challenge the entire literature on oxidase-based biosensors. I recommend the publication of this paper in *eLife* as it stands.

---

## [Author Response]

Essential revisions:The following two main concerns need to be addressed:1) Further characterization and clarification are needed regarding the sensor properties. This is crucial for the potential users in the field to judge and use the sensor, and for interpretation of the biology results using the sensor.

We are grateful to the reviewers and editors to raise such important questions regarding the characterization of sensor properties. The feedback surely contributes to clarify important aspects of the sensor.

i) Clear statement in prominent places about the improvement of the sensor and new potential for its biologic applications separating from the authors' 2015 paper.

Previous enzyme-based biosensor designs, including the ChOx biosensor described in our publication on 2015 (Santos et al., 2015), were based on the differential coating of electrode sites with matrices containing or lacking ChOx. This modification render the sites Chsensitive or insensitive, respectively. The latter have been termed “sentinel” sites, as they are designed to respond to any perturbation except to the analyte of interest (Ch in this case). By subtracting the sentinel from the Ch-measuring site, this approach has been useful to decrease the contribution of interferent signals, namely caused by electrochemical oxidation of electroactive compounds or by voltage fluctuations associated with LFP. However, crosstalk caused by H_2_O_2_ diffusion from enzyme-coated to sentinel sites poses important constraints on this design. The inter-site spacing required to avoid diffusional cross-talk leads, for example, to uncontrolled differences in the amplitude and phase of LFP across sites, compromising common-mode rejection.

In the current study, we have circumvented diffusional cross-talk-related limitations by implementing a novel sensing approach. Rather than changing the coating composition across recording sites, we have differentially modified their electrocatalytic properties towards H_2_O_2_, resulting in Ch-sensitive and pseudo-sentinel sites. As Ch responses depended solely on the intrinsic properties of the metal surface, we could dramatically reduce the size and increase the spatial density of recording sites by using tetrode configuration. Tetrodes, a bundle of four twisted wires glued together, are conventionally used for separating single neuron action potentials based on the spatial structure of their action potentials across wires. Here, the spatial structure of the electrochemical signal is created by electrochemical modification of wires. Importantly this design allows the unbiased measurement of ChOx activity and O_2_ in the same brain spot by using a tetrode site to directly measure the latter. This has not been possible to achieve with conventional enzyme-based biosensor designs, including our own previous stereotrode design.

We acknowledge that the improvements of the TACO sensor over our previous stereotrode design, published in 2015 (as well as other conventional enzyme-based biosensors in general), were not clearly emphasized in the manuscript. We added new paragraphs/sentences in the Introduction and Results of the revised manuscript highlighting the main difference between the two sensors and advantages of the new design for the unbiased measurement of the signals derived from ChOx activity (COA) and O_2_.

ii) Regarding the choline responses: characterizing the linearity of choline response is important for users to understand the sensor properties.

Responses to choline were highly linear within the concentration range tested (up to 30 µM). This information was added to Table 1 and mentioned in the text (Results) of the revised manuscript.

Related, demonstration how to calibrate moving artificial signals in freely-moving rodents will be useful for the future applications.

Movement can cause electromagnetic or mechanical perturbations (movement artifacts) that are expected to scale with the impedance of individual recording sites. As the same applies for LFP-related currents, it is not trivial to discriminate both confounds. Nevertheless, our common-mode rejection approach, which is optimized by a frequency-domain correction of electrode impedances (please check Materials and methods section, for detailed explanation), is designed to optimally remove both LFP- and movement-related artifacts.

In our freely-moving recordings we did not have prominent movement-related perturbations, probably due to the proximity of the head-stage to the sensor and the shielding effect of the grounded copper mesh that covers the implant. Nevertheless, candidate events likely caused by movement consisted in current deflections aligned to locomotion bouts, which were completely removed by common-mode rejection. In the revised manuscript we added the average raw traces triggered on locomotion bouts in Figure 2D, highlighting the usefulness of our method to remove putative movement-related artifacts in addition to LFP and other interferents. We have also added a brief mention to this issue in the Results.

Further, since the COA signal is confounded by phasic O_2_ fluctuations, the authentic changes in COA are potentially interfered by O_2_-evoked enzymatic responses. The interpretation of the signal interference needs to be clearly discussed, including O_2_-evoked changes, and other related signaling changes, like DA.

The main focus of our study was to investigate the effect of physiological O_2_ fluctuations on the ChOx biosensor signal, which is given by the activity of immobilized ChOx, which we abbreviate as COA across the manuscript. In order to address this issue in an unbiased manner it is essential to clean artifacts that directly generate currents on the electrode surface (please see response to point 1vi for details). Our TACO sensor was designed to optimize the removal of such confounds, resulting in a clean COA signal. As this signal reflects the activity of immobilized enzyme, it is sensitive to changes in O_2_, not only Choline. Thus, the COA signal is not confounded, but rather modulated by changes in O_2_. Our main finding was that phasic O_2_ modulation of COA is a major confound of phasic Ch dynamics measurements using ChOx sensors in vivo in the brain. In this sense, the central tenet of the paper is that COA is not reflecting an authentic choline concentration dynamics, but rather a nonlinear function of Ch and O_2_ dynamics, with no feasible analytical approach to separate the two. We recognize that, in the Materials and methods section, the description of how the COA signal was computed could lead to confusion between authentic COA and authentic Ch measurement. In the revised manuscript we have changed the terms used in the signal cleaning procedure.

Regarding neurochemical confounds (e. g. ascorbate or dopamine and other monoamines), we acknowledge that the description of multichannel sensor properties in Table 1 could be confusing to readers. The table was also not conveying the important information on how sensitive is our COA measurement to these artifacts. In the revised manuscript we have removed the information about selectivity ratios for individual sites. Instead, the table section now called “Analytical properties for COA measurement” was expanded and now shows DA and AA sensitivities and selectivity ratios for the COA signal, computed from the difference between Au/Pt/*m*-PD and Au/*m*-PD sites.

Additionally, we added a column in the color plot in Figure 1E describing the relative responses of the COA measurement to the different factors. This addition highlights the high selectivity of the COA signal for Ch, as compared with individual sites.

Finally, we have detailed the interpretation of the freely-moving signals triggered on SWRs and locomotion bouts. In the Materials and methods section of the revised manuscript, we clarify how the differential signals COA_non-mPD_ and NCC (neurochemical confounds) presented in Figure 2 (revised version) were computed. In the description of these results, we also explain how the response patterns of raw and cleaned signals can be used to infer the contribution of different sorts of artifacts, including movement- and LFP-related and those caused by neurochemicals (Results).

iii) The dimensions of the sensor head need to be specified and spelled out clearly. It seems to be around 50 um, but the text seems to suggest 150 um. The individual sensing elements are 17 μm in diameter. If this is true, it is very exciting because it exhibits hemispherical diffusion yielding higher response and enhanced sensitivity. This may improve spatial and temporal resolution if this is in indeed a much smaller sensor as a disk-shaped one.

We thank the reviewers for referring to this point. It is an important detail that was not clearly stated in the manuscript. In the Materials and methods section, the description of the insertion of the tetrode inside a silica tube might have been misleading. In fact, the tetrode actually protrudes 1-2 cm out of the silica tube. This distance assures that the latter is not in contact with the brain in in vivo recordings. The cutting of the twisted ending of the tetrode results in four disc-shaped sensing elements with 17 µm diameter. The total diameter of the tetrode is approximately 60 µm. In the revised manuscript we have clarified and emphasized these details in the Materials and methods section, in the Results and with an additional cartoon in Figure 1A.

iv) The role of the sentinels with differential plating is very interesting, but the function of the sentinels is not clear (Introduction "canceling LFP-related currents"). They consume oxygen. Why does this not result in overlap of the diffusion layer for the choline sensor and therefore affect choline response? Please explain why differential electroplating was employed.

We further clarified the role of the *pseudo*-sentinel sites on the removal of LFP-related currents and neurochemical artifacts and expanded the reasoning behind this approach. Please check the Introduction of the revised manuscript.

When polarized at +0.6 V *vs.* Ag/AgCl, the *pseudo*-sentinel channels display a residual activity towards electrochemical oxidation of H_2_*O_2_*. This electrochemical reaction generates O_2_, but the effect on the local O_2_ concentration is negligible due to the poor sensitivity and very small electrode surface area (17 µm diameter disc). We measured O_2_ (head-fixed mice and in vitro) by electrochemical reduction at -0.2 V *vs.* Ag/AgCl at a *pseudo*-sentinel site (gold-plated without *m*-PD). In this case O_2_ is consumed, but at a very limited extent that does not affect the local O_2_ level in the sensor. In accordance with the expected lack of effect on O_2_ levels, we have confirmed that switching the applied potential on a gold-plated site between +0.6 V and -0.2 V *vs.* Ag/AgCl has no effect on the COA signal. In the revised manuscript we added a supplementary figure (Figure 3—figure supplement 1) describing this observation. Accordingly, we extended the discussion of this topic in the Results section.

v) Please explain how time-dependent behavior of the sensor was measured. This process typically leads to the formation of a film on this electrode surface which can affect sensitivity. According to the authors' 2015 paper, the method for measuring the response time seems rather crude, and may overestimate the response time which is related to the mixing of the solution. This needs to be discussed.

The sensor response times were estimated from the rise of the current in response to analyte additions in a stirred buffer solution, as described in the Materials and methods section. In the revised manuscript, we added a sentence to further clarify the use of this setup to estimate response times. Indeed, this setup is not the most appropriate to precisely determine response times due to the bias introduced by the analyte mixing time after its addition to the buffer. Our previous study (Santos et al., 2015) suggests however that the biggest contribution to the estimated response time is due to diffusion of Ch in the sensor coating. Besides the fact that we cannot precisely determine response times, it is noteworthy that real response times are faster than the values we report. This further highlights the high temporal resolution of the TACO sensor. We added a paragraph discussing this topic in the revised manuscript (Results).

vi) The effect of LFP and other perturbations of sensor responses need to be more clearly explained.

Two main types of artifacts affect the response of enzyme-based electrochemical biosensors: electromagnetic or electrochemical sources that directly generate currents at the electrode surface and biochemical factors that affect the activity of the immobilized enzyme. The first group can be sub-divided into: (a) artifacts that generate faradaic currents, arising from oxidation/reduction of electrochemically active molecules, such as ascorbate or dopamine; (b) artifacts that change the charge distributions at the electrode surface, generating capacitive currents, which in the brain are mainly caused by local fluctuations in field potentials (LFP) generated by the trans-membrane current sources of the surrounding neural tissue. Effectively, LFP causes potential changes at the electrode surface who’s voltage is clamped by the potentiostat circuit, giving rise to apparent current, similar to voltage clamp measurement of the intracellular current. The second group, consisting in biochemical artifacts, comprises mainly the effect of oxygen on enzymatic activity (although other factors such as temperature and pH might have a minor effect, as discussed in the manuscript, Discussion).

Importantly, the strategies devised to reduce artifacts that directly generate electrochemical currents (chemical surface modifications or common-mode rejection) do not control for factors influencing immobilized ChOx activity.

Since O_2_ interference was the main focus of the paper and is thoroughly described throughout the manuscript, in the Introduction of revised manuscript we extended the description of the factors directly generating currents on the electrode surface (Introduction).

2) Re-organization of the manuscript to improve the readability. This manuscript contains the characterization of the TACO sensor and application of this sensor to monitor real-time behavior in freely moving rodents. The design and characterization of the sensor is intermingled with the application of studying the choline biology with the sensor, making the logic flow hard to follow. The arrangement and presentation of the figures need to be improved so readers can appreciate both characterization and applications aspects and how they are tightly linked. This might also involves properly arrange main figures and associated supplementary figures.

We believe this suggestion stems from the expectation that we may have conveyed to the readers regarding the possibility of measuring authentic Ch dynamics in behaving animals with our TACO sensor. Indeed, the TACO sensor design makes it ideally suited for the unbiased measurement of brain Ch dynamics based on ChOx, while controlling for O_2_ changes that might modulate immobilized enzyme activity. However, our data shows that phasic ChOx activity (COA) is dominated by O_2_ fluctuations in the brain of behaving animals. The complexity of the nonlinear interplay between COA and O_2_, which depends on multiple time-scale concentration dynamics of both enzyme substrates made it impossible to extract authentic Ch from the in vivo COA signal.

Following the logic of data presentation in our manuscript, the initial description of TACO sensor design and properties towards COA measurement was followed by its in vivo application in freely-moving and head-fixed rodents, which led to the discovery of the possible O_2_ confound. This, in turn, prompted the next in vivo experiments with causal manipulations to prove the hypothetical confound effect. Next, in vitro experiments were used for more systematic investigation of the details of the confound and its underlying causes guided by the prior in vivo observations. Finally, we used a detailed mathematical model to quantitatively uncover the mechanism of the oxygen confound of the choline-oxidase-based biosensor.

We think this logic of exposition is guiding the reader through our thought process and progresses consistently from the development of novel methodology to evaluation and identification of the confound, and then to unraveling the mechanism in vivo, in vitro and in the model. Reversing the order of presentation would break this logic and hurt the presentation of the story.

We would like to ask the editor for her consent not to follow the suggested major reorganization. Instead, we clarified the internal logic at the end of the Introduction, as well as throughout exposition of the results. Moreover, throughout the revised manuscript we emphasize the focus of our study on phasic COA dynamics instead of putative Ch by replacing terms alluding to the latter by “COA”. Accordingly, we better articulated the motivation for assessing SWR- and locomotion-related signals in freely moving animals (Figure 2) and the interpretation of these results to avoid a biased expectation of the reader that COA signals provide authentic Ch readout. The revised manuscript now provides an unbiased perspective on motivation and interpretation of the in vivo experiments (Results). The bias of COA by O_2_ and the issues associated with derivation of authentic Ch dynamics from our measurements were also further explained in the Discussion. Along the same lines, we have trimmed Figure 2 in order to keep the focus of the paper on phasic dynamics of the COA signal. Namely, we moved panels B and C describing tonic COA dynamics in the original manuscript to a supplementary figure in the revised version (Figure 2—figure supplement 2).